# TLR3 activation enhances abscopal effect of radiotherapy in HCC by promoting tumor ferroptosis

Liman Qiu [1,2,3,4,6], Hongbing Ji[5,6], Kai Wang[5,6], Wenhan Liu[2], Qizhen Huang[5], Xinting Pan[2,3,4], Honghao Ye[2], Zhenli Li[2,3,4], Geng Chen[2,3,4], Xiaohua Xing [2,3,4], Xiuqing Dong[2,3,4], Ruijing Tang [2,3,4], Haipo Xu[2,3,4], Jingfeng Liu[1], Zhixiong Cai [2,3,4✉] & Xiaolong Liu [1,2,3,4✉]

## Abstract

Radiotherapy (RT) has been reported to induce abscopal effect in advanced hepatocellular carcinoma (HCC), but such phenomenon was only observed in sporadic cases. Here, we demonstrated that subcutaneous administration of Toll-like receptor 3 (TLR3) agonist poly(I:C) could strengthen the abscopal effect during RT through activating tumor cell ferroptosis signals in bilateral HCC subcutaneous tumor mouse models, which could be significantly abolished by TLR3 knock-out or ferroptosis inhibitor ferrostatin-1. Moreover, poly(I:C) could promote the presentation of tumor neoantigens by dendritic cells to enhance the recruitment of activated CD8[+] T cells into distant tumor tissues for inducing tumor cell ferroptosis during RT treatment. Finally, the safety and feasibility of combining poly(I:C) with RT for treating advanced HCC patients were further verified in a prospective clinical trial. Thus, enhancing TLR3 signaling activation during RT could provide a novel strategy for strengthening abscopal effect to improve the clinical benefits of advanced HCC patients.

**Keywords** Poly(I:C); Hepatocellular Carcinoma; Radiotherapy; Abscopal Effect; Ferroptosis
**Subject Categories** Cancer; Digestive System; Immunology

## Introduction

Hepatocellular carcinoma (HCC) is the main pathological type of primary liver cancer and the sixth most common cancerous tumor worldwide (Sung et al, 2021). Historically, radiotherapy (RT) is rarely used for treating HCC due to extremely limited clinical outcomes, since the radiation always induces severe liver injury to the cirrhosis liver, such as hepatic failure (Barazzuol et al, 2020; Koay et al, 2018). With the progress of medical radiation technology, precise modern RT is able to only give high-dose of radiation to the tumor while cause negligible damage to the surrounding normal tissues (Koay et al, 2018). Therefore, RT has also been adopted as a palliative local treatment to reduce metastases and relieve symptoms in advanced HCC patients with extrahepatic diseases (He et al, 2019; Jiang et al, 2012; Zhang et al, 2019; Zhou et al, 2014). During radiotherapy, typically, irradiation (IR) and IR-induced ROS could cause DNA damage in tumor cells, which directly leads to tumor cell death or growth stagnation (Gong et al, 2021). Interestingly, IR is also able to induce tumor regression at non-irradiated distant tumor sites in some circumstances, which is called the "abscopal effect" (Arina et al, 2020; Zhang et al, 2022). The abscopal effect is well known as a consequence of the immune response stimulated by RT, which could significantly prolong the survival time of patients with advanced cancer (Ashrafizadeh et al, 2020). Unfortunately, these abscopal effect remains a relatively rare phenomenon, which needs to be further strengthened for improving the clinical benefits of RT.

Mechanistically, tumor cells damaged by local RT could release tumor-associated antigens and increase the expression level of major histocompatibility complex (MHC) class I molecules, which will benefit the uptake and presentation ability of tumor-associated antigens by dendritic cells (DCs), and then induce the infiltration and activation of CD8[+] T cells to attack tumor cells (Ashrafizadeh et al, 2020; Goto, 2019). Moreover, regulated cell death (RCD) pathways including apoptosis, necroptosis, cuproptosis, ferroptosis et al, which were induced by RT, would be crucial factors to induce the abscopal response (Lei et al, 2021; Levy et al, 2016; Zhao and Shao, 2020). However, as far as we know, abscopal response to RT alone in HCC cases was very few (Pangal et al, 2022; Reynders et al, 2015). Such a low incidence of abscopal effects may be due to the increased intratumoral release of immunosuppressive cytokines such as TGFb and CCL2 by IR (Mondini et al, 2020; Zhao and Shao, 2020). Activation of these immunosuppressive signals would render the activated CD8[+] T cells insufficient to maintain long-term positive feedback, ultimately weakening or limiting the effective anti-tumor immune response after RT (Zhao and Shao, 2020). Therefore, appropriate immune modulation is worth to explore for better establishing and maintaining more conducive immune environment to attack tumor cells after RT. To date, RT combined with immune checkpoint inhibitors has been

[1]College of Chemical Engineering, Fuzhou University, Fuzhou, P. R. China. [2]The United Innovation of Mengchao Hepatobiliary Technology Key Laboratory of Fujian Province, Mengchao Hepatobiliary Hospital of Fujian Medical University, Fuzhou, P. R. China. [3]The Liver Center of Fujian Province, Fujian Medical University, Fuzhou, P. R. China. [4]Mengchao Med-X Center, Fuzhou University, Fuzhou, P. R. China. [5]Radiotherapy Department, Mengchao Hepatobiliary Hospital of Fujian Medical University, Fuzhou, P. R. China. [6]These authors contributed equally: Liman Qiu, Hongbing Ji, Kai Wang. ✉E-mail: caizhixiong1985@163.com; xiaoloong.liu@gmail.com

shown to increase the rate of abscopal effect in patients with advanced tumors (Ozpiskin et al, 2019; Zhao and Shao, 2020). However, the rate of enhanced abscopal effects is still unsatisfactory, and a large number of patients are still unable to accept or tolerate it due to the relatively large side effects and expensive drugs. Moreover, the combination of RT with IL-2 has been shown to be well tolerated and immunologically active in metastatic melanoma and renal cell carcinoma in clinical trials, even if its efficacy relative to RT alone is still not obvious (Bulgarelli et al, 2021; Curti et al, 2020; Hannan et al, 2021).

Polyinosinic acid-polycytidylic acid [poly(I:C)] is a double-stranded RNA (dsRNA) synthetic analog, consisting of a chain of double-stranded inosine (I) and cytidine (C). As a TLR3 agonist, poly(I:C) could strongly induce the production of multiple interferon (IFN) and enhance several immune responses with different mechanisms, including DC priming and antigen presentation, as well as other auxiliary signals of immune activation (Le Naour et al, 2020; Matsumoto and Seya, 2008). In clinical, poly(I:C) injection has been used as the adjuvant treatment for viral keratitis, herpes simplex infection, and chronic viral hepatitis in China. In our previous clinical trial, poly(I:C) injection could serve as an immune adjuvant for tumor neoantigen vaccine in HCC patients, and support the long-acting adjuvant effect in tumor neoantigen presentation (Cai et al, 2021). Although the immune responses activated by RT is conducive to establish tumor-specific toxicity, the activated immune state lacks sufficient durability and potency. Therefore, combination of poly(I:C) with RT may provide an appropriate strategy to promote the abscopal effect during RT. A phase I clinical trial by the combination of intra-tumoral injection of the TLR3 agonist poly-ICLC after local radiation at relatively low dose (2.5 Gy/fraction, a total dose of 22.5 Gy) and local regional treatment (TAE or TACE) to treat advanced HCC patients, has proved to be safe and tolerable (de la Torre et al, 2017). However, the abscopal effect has been only observed in a small number of patients and the underlying molecular mechanism remains unclear (de la Torre et al, 2017). Hence, it is extremely necessary to further enhance the potential efficacy and explore molecular mechanisms of such combination in preclinical animal models and clinical studies.

In this study, the combined treatment plan of RT and poly(I:C) was developed and optimized in a noninvasive manner by using therapeutic dose of radiation for targeted HCC lesion and injecting poly(I:C) subcutaneously during RT treatment. Firstly, the bilateral HCC subcutaneous tumor mouse models were constructed to explore the effect of poly(I:C) on abscopal effect during RT and systematically investigated its underlying molecular mechanisms. Furthermore, we also conducted an investigator initiated prospective clinical trial by combining poly(I:C) with RT to confirm the safe and feasibility of RT combined with poly(I:C) in enhancing the abscopal effect in advanced HCC patients. This study will clarify the molecular mechanism of TLR3 signaling activation in the abscopal effect of RT for HCC, and provide feasible clinical strategies for improving the clinical benefits of RT.

# Results

## Poly(I:C) enhances the abscopal effect of radiotherapy in HCC mouse model

To further explore the anti-tumor effect and potential mechanisms of the combined treatment of poly(I:C) and RT, the mouse HCC

cell line Hepa1-6 cells were subcutaneously injected into the left hind limb (indicated as Tumor1: irradiated tumor) and right armpit (indicated as Tumor2: non-irradiated tumor) to construct bilateral HCC subcutaneous tumor mouse model to mimic the tumor metastasis of advanced HCC. Then the bilateral HCC subcutaneous tumor mouse models were randomly divided into 4 groups and further received normal saline (NS) treatment, RT treatment alone [total 40 Gy irradiated in Tumor1 (8 Gy/fraction)], poly(I:C) treatment alone [200 μl (1 mg/ml), injection subcutaneously] and poly(I:C) plus RT treatment as indicated in Fig. 1A. The irradiation area (Tumor1) of mice treated with RT is shown in Fig. 1B. After 14 days from receiving RT treatment, the grow rate of Tumor1 was significant inhibited in the treatment groups of RT alone and poly(I:C) plus RT, but showed obviously progression in both treatment groups of NS and poly(I:C) alone (Fig. 1C,D). Meanwhile, growth of Tumor1 in mice treated with poly(I:C) plus RT was markedly slower than that of mice treated with RT alone ($P = 0.024$, Fig. 1D, upper panel). Significantly, the inhibition of tumor progression in Tumor2, which were defined as abscopal effect of RT, was only shown in poly(I:C) plus RT group when compared with RT treatment alone ($P = 0.002$, Fig. 1D, bottom panel). Moreover, the Kaplan-Meier analysis of distant tumor progression demonstrated the longer progression-free of the mice with poly(I:C) plus RT treatment when compared with RT treatment alone ($P = 0.019$, Fig. 1E). These results suggested that subcutaneous administration of poly(I:C) during RT treatment could not only inhibit the growth of irradiated tumor, but also could promote the abscopal effects in irradiated HCC mice, which further improved the prognosis of HCC bearing mice treated with RT.

## Poly(I:C) promotes tumor ferroptosis in irradiated HCC mice

Given that irradiation induces various regulatory cell death (RCD) of tumor cells (Lei et al, 2021), we attempted to investigate whether poly(I:C) could promote the RCD efficiency induced by RT. Firstly, transcriptome sequencing of Tumor1 and Tumor2, which were collected from HCC mice with different treatment as indicated, was performed to analysis the degree of RCD. As shown in Fig. 2A, RCD enrichment score analysis revealed that only ferroptosis score was significantly enriched in both Tumor1 and Tumor2 collected from mice treated with poly(I:C) plus RT when compared with RT treatment alone. To further confirm the enhanced ferroptosis signal in poly(I:C) plus RT treatment, ferroptosis associated biochemical characteristics [accumulation of lipid peroxides malondialdehyde (MDA) and glutathione (GSH) depletion] were evaluated in Tumor1 and Tumor2. As shown in Fig. 2B,C, there were no significant changes of MDA accumulation and GSH consumption in mice treated with poly(I:C) alone comparing with that in mice treated with NS. In contrast, RT alone significantly increased the MDA accumulation and GSH consumption in Tumor1 (MDA content: NS $0.97 \pm 0.07$ vs RT $1.34 \pm 0.06$, $P = 0.015$; GSH content: NS $164.37 \pm 17.55$ vs RT $103.75 \pm 6.82$, $P = 0.018$), but not in Tumor2, indicating that the RT-induced ferroptosis-related events were only restricted to irradiated tumor tissues (Fig. 2B,C). Significantly, poly(I:C) plus RT treatment not only promoted the RT-induced MDA accumulation [RT $1.34 \pm 0.06$ vs Poly(I:C) + RT $1.69 \pm 0.05$, $P = 0.018$] and GHS consumption [RT $103.75 \pm 6.82$ vs

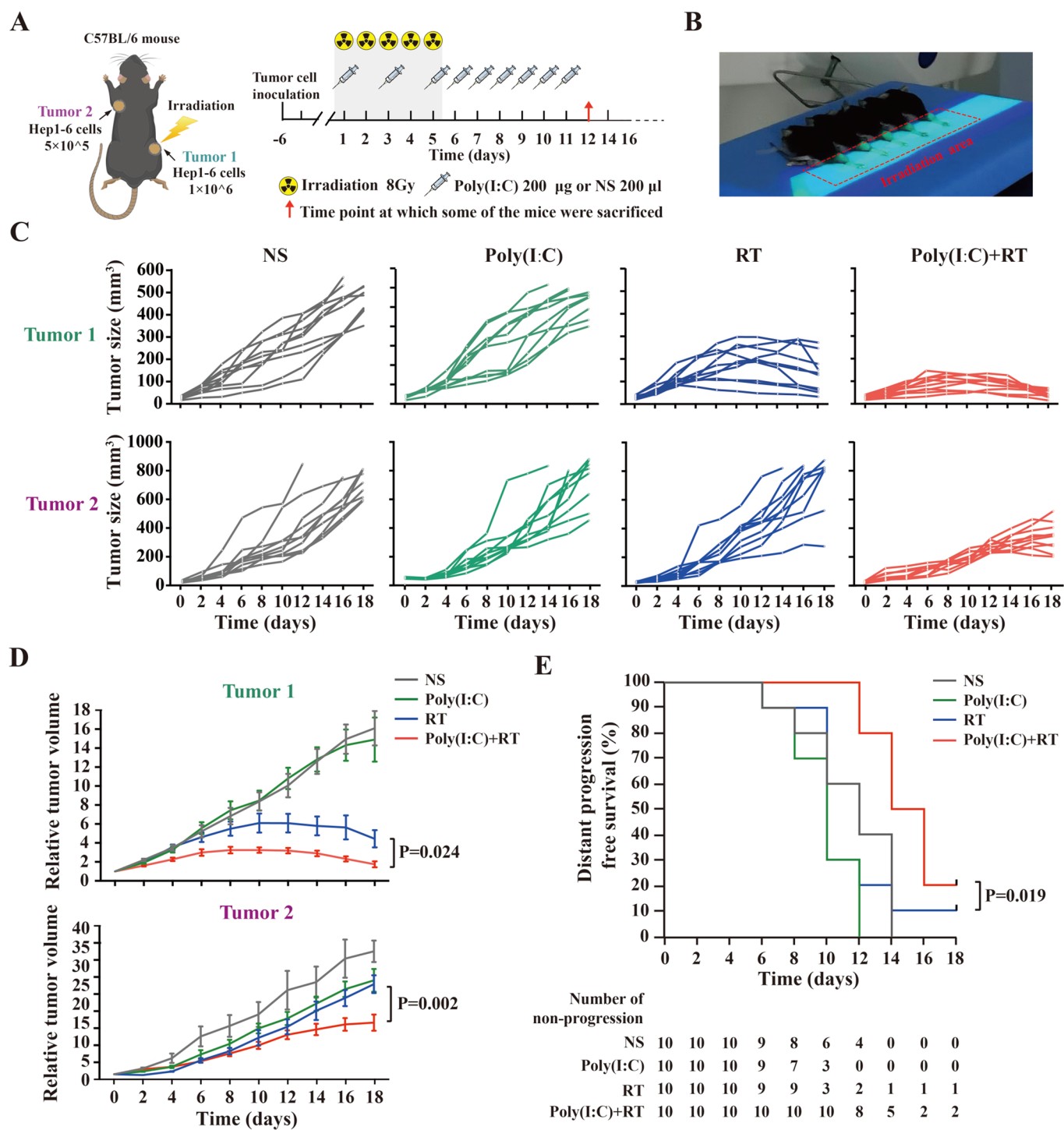

**Figure 1. Poly(I:C) enhances the abscopal effect of radiotherapy in HCC mouse model.**

(A) Schematic diagram for mouse subcutaneous HCC tumor modeling and therapy. (B) The tumors in the hind limbs of mice were fixed on the flat board and exposed to the irradiation. (C) Tumor growth curves of directly irradiated tumor (Tumor1) and distant tumor (Tumor2) in different treated group ($n = 10$ mice for each group, i.e., two biological replicates of $n = 5$ mice per group) as indicated. (D) ANOVA was used to analyze the difference in tumor volume in different treated groups ($n = 10$ mice for each group, i.e., two biological replicates of $n = 5$ mice per group). Results are shown as mean ± SEM (error bar). Relative tumor volume of Tumor1 and Tumor2 in different treated group was shown as indicated. (E) The difference of distant progression survival in different treated group ($n = 10$ mice for each group, i.e., two biological replicates of $n = 5$ mice per group) was analyzed by Kaplan–Meier algorithm, and the Kaplan-Meier curve was shown as indicated. The abscopal tumor (Tumor2) reaching 300 mm³ was defined as distant progression. The experiment was repeated twice, each time with 5 mice per group (a total of 10 mice in each group). Results are shown as mean ± SEM (error bar). Source data are available online for this figure.

Poly(I:C) + RT $51.51 \pm 6.36$, $P = 0.046$] in Tumor1, but also induced MDA accumulation [RT $1.18 \pm 0.09$ vs Poly(I:C) + RT $1.81 \pm 0.15$, $P = 0.026$] and GSH consumption [RT $113.06 \pm 3.23$ vs Poly(I:C) + RT $69.42 \pm 6.73$, $P = 0.027$] in Tumor2 (Fig. 2B,C). Meanwhile, the apoptosis and necrosis in either Tumor1 or Tumor2 show no significant change when treated with poly(I:C) plus RT compared to RT treatment alone (Appendix Fig. S1). Taken together, these findings suggested that poly(I:C) could enhance the RT-induced tumor ferroptosis in irradiated tumors and non-irradiated distant tumors.

To further investigate whether poly(I:C) improves the therapeutic efficiency of radiation by enhancing tumor ferroptosis, the mice treated with poly(I:C) plus RT was intravenously injected with ferroptosis inhibitor Fer-1 (Fig. 2D). As shown in Fig. 2E, the tumor burden of Tumor1 and Tumor2 in all mice treated with RT plus poly(I:C) were both showed tumor regression. This result further confirmed the abscopal effect of combinational therapy in HCC treatment. However, further injecting with Fer-1 could obviously abolish the growth inhibitory effect of poly(I:C) plus RT treatment on Tumor1 and Tumor2 [Tumor1 volume: Poly(I:C) + RT $8.92 \pm 3.56$ vs Poly(I:C) + RT+Fer-1 $116.28 \pm 29.74$, $P < 0.001$; and Tumor2 volume: Poly(I:C) + RT $142.20 \pm 16.93$ vs Poly(I:C) + RT+Fer-1 $406.32 \pm 58.19$, $P < 0.001$]. Further detection of MDA level and GSH consumption in the Tumor1 and Tumor2 revealed that Fer-1 could reverse poly(I:C) plus RT treatment-induced increase in MDA accumulation (From $2.19 \pm 0.13$ to $1.72 \pm 0.04$ in Tumor1, $P = 0.004$; from $1.98 \pm 0.08$ to $1.56 \pm 0.09$ in Tumor2, $P = 0.003$; Fig. 2F, left panel) and GSH consumption (From $62.83 \pm 6.37$ to $121.22 \pm 4.31$ in Tumor1, $P < 0.001$; from $69.71 \pm 8.53$ to $120.56 \pm 3.42$ in Tumor2, $P = 0.001$; Fig. 2F, right panel), respectively. In addition, since RT treatment itself generates ROS at the tumor site, it can also induce ferroptosis in tumor cells, leading to potential tumor control. To evaluate the role of RT-induced ferroptosis and ROS in tumor control during poly(I:C) plus RT treatment, Fer-1 was added to the HCC mouse model receiving RT alone, and ROS inhibitor NAC was added to the HCC mouse model receiving poly(I:C) plus RT treatment, respectively. As shown in Fig. EV1, neither the addition of Fer-1 to RT treatment alone nor the addition of ROS inhibitor NAC to poly(I:C) plus RT treatment significantly altered the tumor control and ferroptosis-related events. Furthermore, to further verify the abscopal effect induced by poly(I:C) plus RT and associated mechanisms, BALB/c mouse subcutaneous tumor model constructed by H22 cell line was treated with poly(I:C) plus RT with or without Fer-1/NAC. As expected, these phenomena were similar with the results found in C57BL/6 mouse subcutaneous tumor model constructed by Hepa1-6 cell line (Fig. EV2). Taken together, these results suggested that poly(I:C) could enhance radiotherapy efficacy in HCC by promoting tumor ferroptosis at both irradiated or non-irradiated tumor sites, which are not directly depending on RT-induced tumor ferroptosis or ROS generation.

## Poly(I:C) enhances RT induced abscopal effect depending on TLR3 signaling

Toll-like receptor 3 (TLR3) is one member of the Toll-like receptor family, which can recognize foreign double-stranded RNA, induce the activation of NF-kB and the production of cytokines. Poly(I:C), recognized by TLR3, could promote the maturation and activation of dendritic cells, and enhance the activity of cytotoxic T cells, etc., thereby promoting the immune system to enhance the ability to recognize and attack tumor cells. Therefore, to clarify the role of TLR3 signaling in poly(I:C)-enhanced abscopal effect of radiotherapy, TRL3-knockout mice (Homozygous TLR3$^{-/-}$) were used to construct HCC subcutaneous tumor model, and further receiving poly(I:C) plus RT treatment (Fig. 3A). As shown in Fig. 3B,C, the abscopal effect, as well as ferroptosis-related events in Tumor2 from TRL3-knockout mice receiving poly(I:C) plus RT treatment were vanished, indicating that TLR3 signaling activation is indispensable for poly(I:C) to promote abscopal effect of RT.

Moreover, to identify whether poly(I:C) promotes tumor ferroptosis directly acting on tumor cells, the Hepa1-6 cells treated with or without RT were further received different concentrations (0, 1, 5, and 10 μg/ml) of poly(I:C) treatment in vitro. The CCK-8 assay showed that there is no obvious toxicity to tumor cells when the concentration of poly(I:C) reaches 10 μg/ml (high concentration) in medium. Further combined with different doses of RT (4 Gy and 8 Gy) in vitro to treat Hepa1-6 cells, no obvious synergistic therapeutic effect was found (Fig. 3D). Collectively, our results suggest that enhancing the tumor ferroptosis of RT by poly(I:C) in vivo might do not directly act on tumor cells.

## Poly(I:C) promotes DC uptake of neoantigens released from irradiated-tumor tissues

Since RT could promote the release of tumor-associated antigens (TAAs)/neoantigens in vivo (Lussier et al, 2021), poly(I:C) is also recognized to enhance the ability of antigen uptake by activating TLR3 signaling pathway (De Waele et al, 2021). Therefore, the effect of poly(I:C) during RT on the uptake and presentation of TAAs or neoantigens at the directly irradiated Tumor1 was further investigated. Since no definite TAAs have been identified in Hepa1-6 cell line, here we intend to use 7 neoantigens with strong immunogenicity that we have previously identified in Hepa1-6 cell line to evaluate neoantigen-specific immune response induced by poly(I:C) plus RT (Chen et al, 2022). As shown in Fig. 4A, an obvious neoantigen-specific immune response was observed in HCC mouse model treated with poly(I:C) plus RT, but not in NS treatment, poly(I:C) treatment alone and RT treatment alone. Given that poly(I:C) could promote antigen uptake and presentation by DCs, and thereby promoting maturation of DCs (De Waele et al, 2021), the effect of poly(I:C) on neoantigen presentation was further investigated by examining the maturity of DCs in mouse models. Firstly, flow cytometry was used to assess the maturation status of DCs from lymph nodes in HCC mouse models received different treatments as indicated. As shown in Fig. 4B, poly(I:C) combined with RT treatment could significantly improve the maturity of DC in lymph nodes when compared with other groups [Poly(I:C) + RT $48.32 \pm 0.86\%$ vs NS $32.21 \pm 2.37\%$, poly(I:C) alone $36.45 \pm 2.75\%$, or RT alone $37.41 \pm 3.00\%$; $P < 0.05$].

Notably, conventional type 1 dendritic cells (cDC1s) have been shown to be essential for the efficacy of cancer immunotherapy (Liang et al, 2021; Schenkel et al, 2021). In particularly, cDC1s are thought to be presented in tumor microenvironment to uptake tumor-associated antigens and potently cross-prime initiating of tumor-responsive CD8$^+$ T cells (Blair et al, 2020). Therefore, we further analyzed the number of cDC1s (marked with CD45$^+$CD11c$^+$CLEC9$^+$) in Tumor1 by multi-color

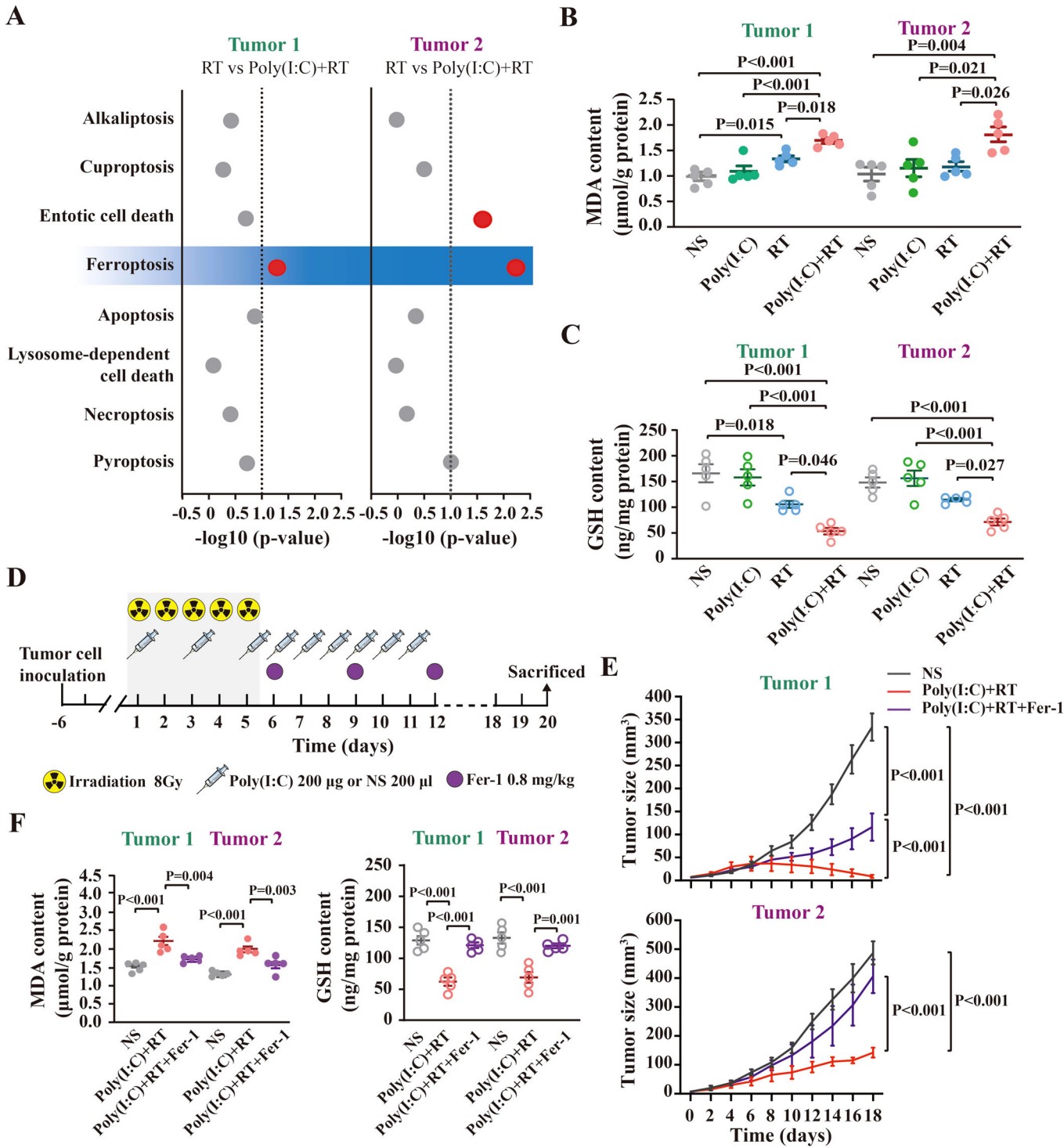

**Figure 2.  Poly(I:C) promotes tumor ferroptosis in irradiated HCC mice.**

(A) *P* value (analyzed by t-test) of RCD enrichment score in Tumor1 and Tumor2 between mice treated with RT and poly(I:C) plus RT. Determination of MDA content (B) and GSH content (C) in Tumor1 and Tumor2 in mice with different treatment ($n = 5$ independent samples for each group) as indicated (illustrated by scatterplot). The dada are shown as mean ± SEM (error bar), and ANOVA was performed to analyze the differences among groups. (D) Schematic diagram of ferroptosis inhibitor Fer-1 treatment in HCC mouse model receiving poly(I:C) plus RT. (E) Tumor growth curves of Tumor1 and Tumor2 in different treated group ($n = 5$ mice for each group) as indicated. Results are shown as mean ± SEM (error bar). (F) Determination of MDA content (left panel) and GSH content (right panel) in Tumor1 and Tumor2 in different treated group ($n = 5$ independent samples for each group) as indicated (illustrated by scatterplot). The dada are shown as mean ± SEM (error bar), and ANOVA was performed to analyze the differences among groups. Source data are available online for this figure.

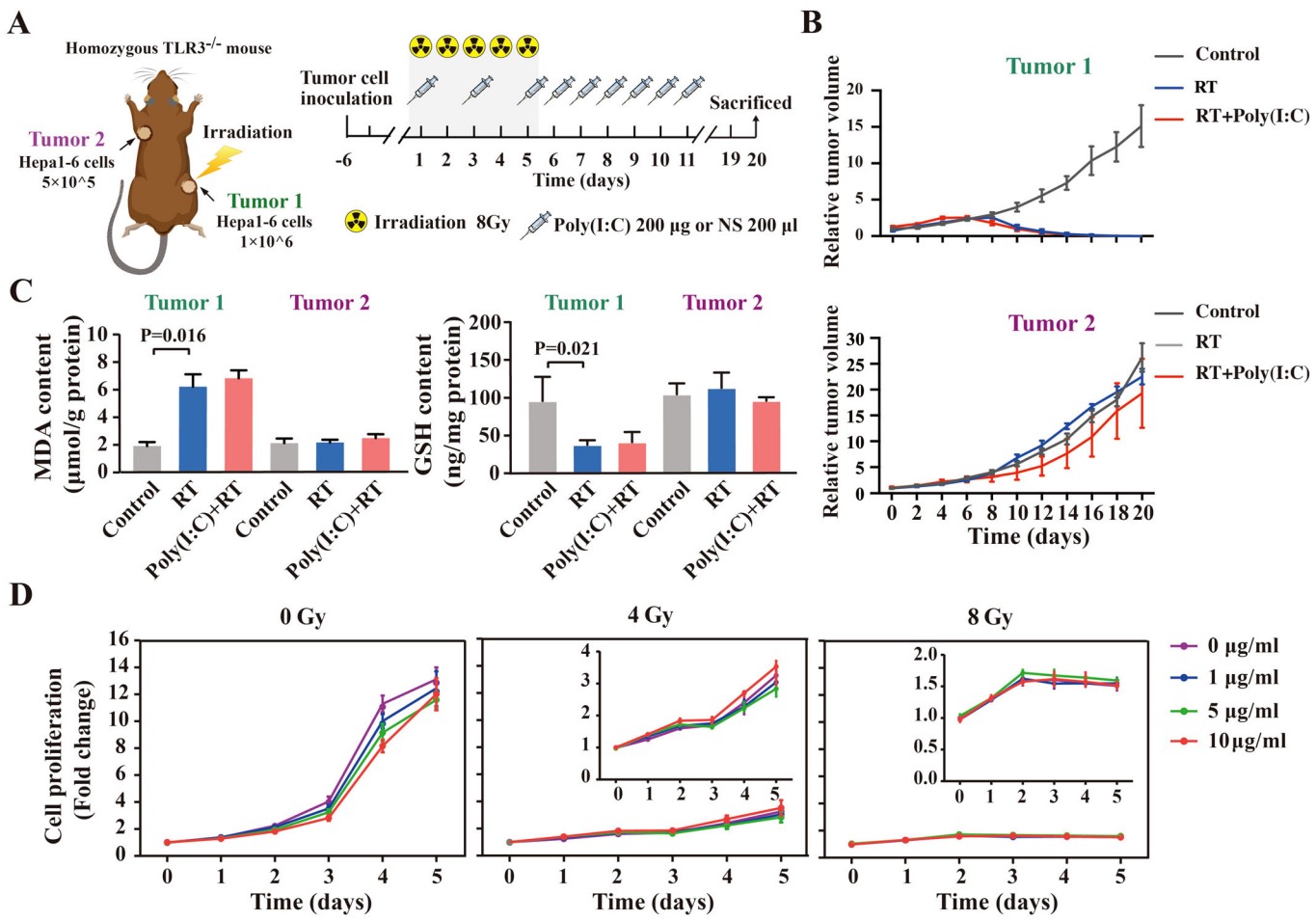

**Figure 3. Poly(I:C) enhances RT induced abscopal effect depending on TLR3 signaling.**

(A) Schematic diagram for TLR3-knockout mouse subcutaneous HCC tumor modeling and therapy. (B) Relative tumor volume of Tumor1 (left panel) and Tumor2 (right panel) in different treated groups as indicated (*n* = 4 mice for each group). Results are shown as mean ± SEM (error bar). (C) Determination of MDA content (left panel) and GSH content (right panel) in different treated groups (*n* = 3 mice in Control group, *n* = 4 mice in RT group, *n* = 4 mice in Poly(I:C) + RT group) as indicated (illustrated by scatterplot). The data represent the means ± SEM (error bar), and the differences among groups were analyzed by ANOVA. (D) Proliferation assessment of Hepa1-6 cells stimulated with different content of poly(I:C) under different doses of irradiation by CCK-8 assay. To better visualize the proliferation of Hepa1-6 cells under 4 Gy and 8 Gy irradiation, the cell proliferation curves that were adjusted to the optimal vertical coordinates were shown as inset graphs in the upper right corner. The cell proliferation of cell growth in poly(I:C) of 0 µg/ml group for 1 day was normalized to 1. The data represent the means ± SEM (error bar) of at least three independent experiments. Source data are available online for this figure.

immunofluorescence, and the results showed the cell density of cDC1s in mice treated with poly(I:C) plus RT was 2.84-fold higher than that in mice treated with RT alone [RT 645.03 ± 71.47 vs Poly(I:C) + RT 1833.18 ± 177.46; *P* < 0.001, Fig. 4C]. Moreover, poly(I:C) combined with RT significantly increased the IL-12 (cytokine associated with enhanced DC priming) expression of intratumoral cDC1s, compared with RT alone [IL12+ cDC1s: RT 352.63 ± 27.92 vs Poly(I:C) + RT 743.51 ± 74.32; *P* < 0.001] (Fig. 4C). Moreover, we also noted that the proportion of CD8+ central memory T cells (CD8+CD44+CD62L+) in the spleens of mice treated with poly(I:C) plus RT was significantly higher than that of mice treated with poly(I:C) alone or RT alone (Fig. 4D), suggesting that poly(I:C) maintains the response of T cells to specific antigens in mice treated with RT. Overall, poly(I:C) significantly promoted the uptake and presentation of neoantigens released from irradiation-damaged tumors by DCs, and effectively

induced the generation of specific tumor-reactive T cells and central memory CD8+ T cells.

## Poly(I:C) promotes T cell infiltration in tumor tissues during RT

To evaluate the role of poly(I:C) on tumor microenvironment during RT, ImmuCellAI was used to analyze the transcriptome sequencing data of tumor tissues from mice with different treatments as indicated for generating infiltration score of multiple immune cells. As shown in Fig. 5A, T cell infiltration score of T cell subsets in both Tumor1 and Tumor2 were reduced by RT as a whole, while poly(I:C) plus RT could significantly improve infiltration score of T cells, especially CD8+ T cells, in both Tumor1 and Tumor2 of HCC tumor-bearing mice [Tumor1: RT 0.016 ± 0.005 vs Poly(I:C) + RT 0.118 ± 0.016, *P* = 0.005; Tumor2:

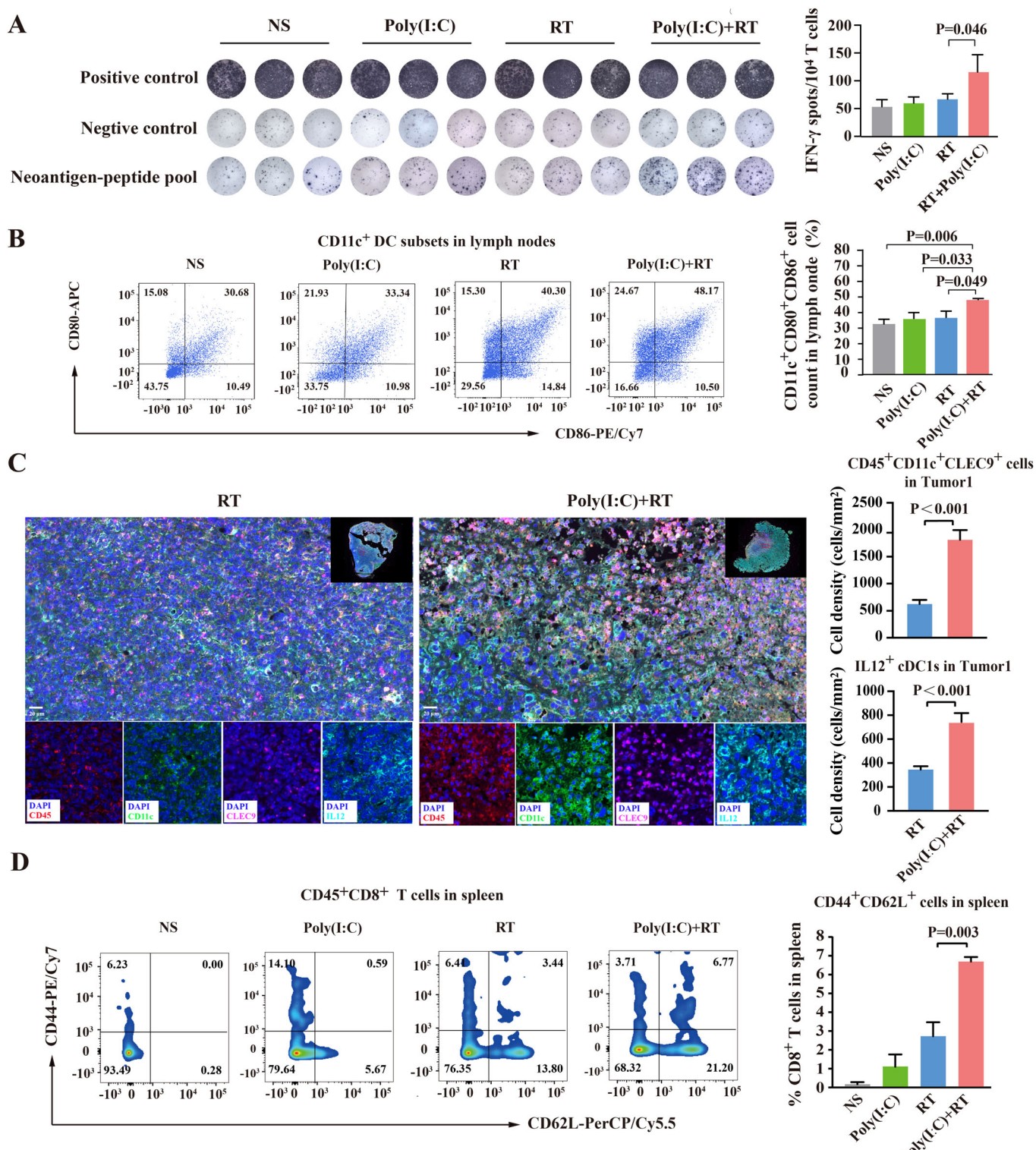

RT 0.015 ± 0.005 vs Poly(I:C) + RT 0.079 ± 0.013, $P = 0.010$]. This phenomenon was further confirmed by immunohistochemical staining of CD8 and granzyme B (GzmB) (Figs. 5B and EV3), as well as Elisa detection of the multiple inflammatory factors (TNF-α, IFN-γ, IL-2, IL-6, IL-12) in Tumor1 and Tumor2 of HCC tumor-

bearing mice with Poly(I:C) plus RT treatment (Fig. 5C). However, poly(I:C) did not significantly affect the intratumoral infiltration of macrophages and regulatory T cells (Treg), which were highly sensitive to oxidized lipids (Appendix Fig. S2). Moreover, GO&Pathway analysis also revealed that poly(I:C) plus RT

**Figure 4. Poly(I:C) promotes DC uptake of neoantigens released from irradiated-tumor tissues.**

(A) ELISpot assay showing neoantigen specific-reactivity of splenic T cells against neoantigen peptide-pool with 7 mutant neoantigens identified in HCC Hepa1-6 cell line, and the results of three replicates were shown in statistical bar chart (right panel). The differences among the groups were analyzed by ANOVA. (B) The proportion of CD80+CD86+ cells (gated on CD11c) in Tumor1 of different treated group as indicated was shown by representative flow cytometry scatter plots (left panel). The results of three replicates were shown in statistical scatter plots (right panel), and the differences among groups were analyzed by ANOVA. (C) The representative immunofluorescence image of CD45+CD11c+CLEC9+ cDC1s in Tumor1 of different treated group (left panel), and the cell density of cDC1s (CD45+CD11c+CLEC9+ cells) and IL12+ cDC1s of entire section field of the tumor tissue samples were shown in statistical scatter plots (n = 3 slides from different mouse per group, right panel). Scale bars, 20 μm (400×). The differences between two groups were analyzed by t-test. (D) Flow cytometry showing the percentage of CD8+ central memory T cells in spleen from different treated group as indicated (left panel). The results of three replicates were shown in statistical scatter plots (right panel), and the differences among groups were analyzed by ANOVA. Data information: In (A–D), statistical data were presented as mean ± SEM (error bar) of three independent experiments. Source data are available online for this figure.

treatment resulted in significant enrichment of oxidative phosphorylation signaling pathway and cellular mitochondrial structure-related cellular components (mitochondria, as the primary regulator of oxidative phosphorylation, could regulate the sensitization of cells to ferroptosis (Wang et al, 2020; Bock and Tait, 2020) in both Tumors (especially in Tumor2), when compared with RT alone (Fig. EV4). Taken together, these results suggested that poly(I:C) could promote the recruitment of immune cells to the irradiated tumor site and unirradiated tumor site, and improve the sensitization of tumor cells to ferroptosis.

Previous studies have confirmed that vascular endothelial cells could secrete adhesion molecules (such as *VCAM-1*) and induce the accumulation of chemokines on the surface of vascular endothelial cells in the inflammatory response, thereby driving T cells to infiltrate into tumor tissues (Vilgelm and Richmond, 2019). Therefore, we further detected the expression of adhesion molecules and chemokines in vascular endothelial cells in Tumor1 and Tumor2 to clarify whether poly(I:C) also increases the recruitment ability of vascular endothelial cells in tumor tissue to immune cells after RT. The expression of *VCAM-1* and multiple chemokines (*CXCL10, CCL2, CCL5, CXCL9, CCL11, CCL8, CCL3, CCL4*) in vascular endothelium sorted from Tumor1 and Tumor2 were further analyzed by qRT-PCR. Unsurprisingly, when compared with RT treatment, poly(I:C) plus RT treatment group significantly up-regulated the expression of *VCAM-1* (6-fold in Tumor1, $P < 0.001$; and 2-fold in Tumor2, $P = 0.014$), and the expression of multiple chemokines (2.0–6.7-fold) in Tumor1 and Tumor2, respectively (Appendix Fig. S3). These results revealed that poly(I:C) promotes the high expression of chemokines in tumor vascular endothelial cells, providing support for immune cell recruitment and inflammatory response in distant tumor sites, and thus promoting the abscopal effect of RT.

## The promoting effect of poly(I:C) on irradiation-induced tumor ferroptosis is dependent on the activation of TLR3 signaling of tissue-infiltrating CD8+ T cells

Recent studies have reported that immunotherapy-activated CD8+ T cells could promote tumor cell lipid peroxidation to enhance the degree of tumor ferroptosis (Wang et al, 2019). Likewise, we also found that poly(I:C) can promote TLR3+CD8+ T cell infiltration to both Tumor1 and Tumor2 when combined with RT (Fig. 5B, Tumor1: RT 2.40 ± 0.73 vs Poly(I:C) + RT 17.22 ± 2.58, $P < 0.001$; Tumor2: RT 2.43 ± 0.36 vs Poly(I:C) + RT 7.75 ± 0.80, $P < 0.001$). However, whether poly(I:C) induces tumor cell ferroptosis through CD8+ T cells to exert abscopal effects remains unclear. To clarify the role of poly(I:C)-activated intratumoral infiltrating CD8+ T cells in abscopal effect, CD8+ T cells derived from Tumor2 of

bilateral HCC subcutaneous tumor models constructed in C57BL/6 mice or TLR3-knockout mice were both stimulated with PBS or poly(I:C), respectively, and further co-cultured with non-irradiated Hepa1-6 cells to mimic the recruitment of CD8+ T cells into the non-irradiated distant tumor (Fig. 6A). As shown in Fig. 6B upper panel, comparing with PBS treatment, poly(I:C) stimulation significantly increased the inhibition rate of Hepa1-6 cell proliferation by CD8+ T cells with wild-type TLR3 gene (CD8+TLR3wt T cells) derived from HCC subcutaneous C57BL/6 mouse model ($P = 0.012$). However, this phenomenon was not observed using CD8+ TLR3-/- T cells from HCC subcutaneous TLR3-knockout model.

Meanwhile, the ferroptosis of tumor cells detected by the ferroptosis-sensitive lipid peroxidation probe C11-BODIPY using flow cytometry also comfirmed that poly(I:C) stimulated CD8+TLR3wt T cells, rather than CD8+TLR3-/- T cells, could significantly induce remarkable more tumor ferroptosis than the PBS-treated CD8+ T cells ($P < 0.001$, Fig. 6B, bottom panel). More interestingly, when Fer-1 was further added, the anti-tumor killing ability and ferroptosis induced by poly(I:C) stimulated CD8+TLR3wt T cells were reversed ($P < 0.001$, Fig. 6B). These results verified that poly(I:C)-activated intratumoral CD8+ T cells, but not poly(I:C) itself, promoted the ferroptosis-related lipid peroxidation damage and tumor killing in non-irradiated Hepa1-6 cells, suggesting the key role of intratumoral infiltrating CD8+ T cells in poly(I:C) promoted distant tumor ferroptosis and abscopal effect depending on the activation of TLR3 signaling. To further detect whether CD8+ T cells can also induce ferroptosis in tumor cells by secreting some cytokines, the culture supernatant of poly(I:C) or PBS-stimulated CD8+TLR3wt T cells were added to co-incubate with Hepa1-6 cells. As shown in Fig. 6C, when compared with the supernatant of control or PBS simulated CD8+TLR3wt T cells, the supernatant of poly(I:C)-stimulated CD8+TLR3wt T cells could significantly inhibit the growth of Hepa1-6 cells ($P < 0.001$); however, when Fer-1 was added, the relevant inhibitory effect was removed (Fig. 6C, upper panel). The same phenomenon was also observed in lipid peroxidation (Fig. 6C, middle panel) and GSH depletion (Fig. 6C, bottom panel). Further detection of the culture supernatant of CD8+TLR3wt T cells by mass spectrometry found that poly(I:C)-stimulated CD8+TLR3wt T cells could specifically secrete some ferroptosis associated proteins (such as GOT1, and CTSB) to induce tumor ferroptosis when compared with PBS simulated CD8+TLR3wt T cells (Dataset EV2).

To further confirm that CD8+ T cells play a key role in poly(I:C)-induced ferroptosis in vivo, CD8-blocking antibody were intraperitoneally injected into bilateral HCC subcutaneous C57BL/6 mouse model treated with poly(I:C) plus RT for deleting CD8+

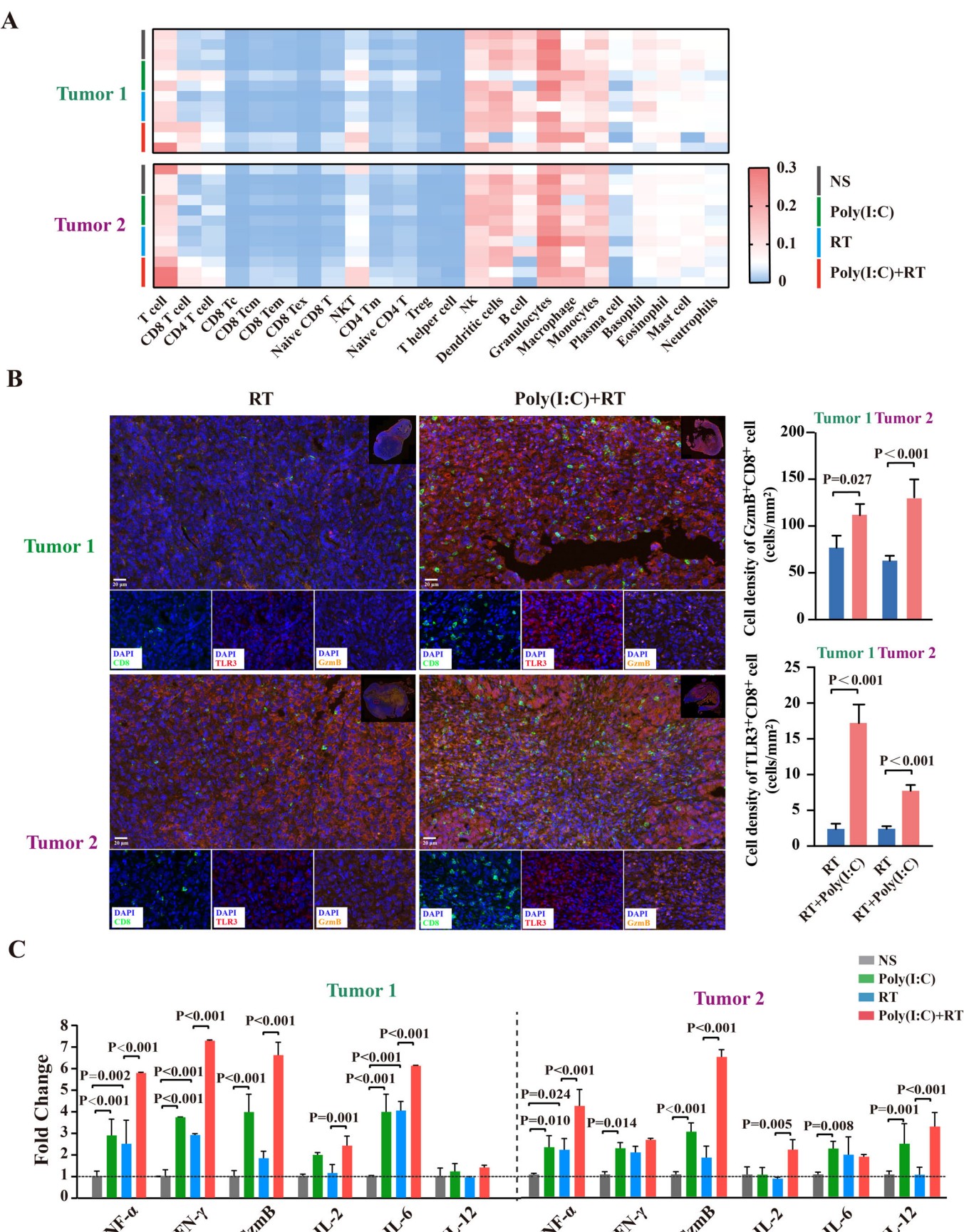

**Figure 5. Poly(I:C) promotes T cell infiltration in tumor tissues.**

(A) Infiltration score of immune cells in tumors tissues according to RNA sequencing data analyzed by ImmuCellAI. (B) The representative immunofluorescence image of TLR3⁺CD8⁺ T cells and GzmB⁺CD8⁺ T cells in tumor tissues (left panel), and the cell density of TLR3⁺CD8⁺ T cells and GzmB⁺CD8⁺ T cells were shown in statistical scatter plots ($n = 3$ slides from different mouse per group, right panel). Scale bars, 20 μm (400×). The cell density are represented by mean ± SEM(error bar) of entire section field of the tumor tissue samples, and the differences between two groups were analyzed by t-test. (C) The expression of TNF-α, IFN-γ, GzmB, IL-2, IL-6, and IL-12 in tumor tissues was detected by Elisa. The expression level in NS group was normalized to 1. Data were expressed as mean ± SEM (error bar) of three independent replicates, and the differences among groups were analyzed by ANOVA. Source data are available online for this figure.

T cells in vivo (Fig. 6D). As expected, when CD8⁺ T cells were deleted in vivo, the abscopal effect (decrease in Tumor2 volume) induced by the combined treatment was significantly attenuated ($P = 0.018$); meanwhile, the regression of irradiated tumors was also attenuated ($P = 0.027$) (Fig. 6E; Appendix Fig. S4). Furthermore, as shown in Fig. 6F, ferroptosis associated MDA accumulation and GSH consumption in Tumor1 and Tumor2 induced by poly(I:C) plus RT treatment were both significantly abolished by CD8⁺ T cell deletion in vivo. These phenomena suggested that poly(I:C) may be partially dependent on CD8⁺ T cells to induce abscopal effects after RT treatment. In summary, all the above results elucidated that poly(I:C) could improve RT-induced ferroptosis in RT targeted tumor and distant tumor by promoting the tumor antigen presentation ability of DC and stimulating the TLR3 activated CD8⁺ T cell infiltration in tumor tissues to secrete ferroptosis-related factors, thus increasing the frequency and potency of abscopal effect of RT in HCC (Fig. 6G).

## Poly(I:C) improves radiotherapy efficacy in patients with advanced HCC

To further evaluate the role of poly(I:C) for enhancing the potency of irradiation on direct tumor destruction and promoting the abscopal effect of RT in patients with advanced HCC, we conducted a single-center, single-arm, open-label clinical trial of combining poly(I:C) injection with RT in the treatment of advanced HCC patients (registration number: ChiCTR2100053441). In brief, the enrolled patients firstly received RT treatment to the target tumor lesion according to established standard plan by experienced doctors, and subcutaneously injected poly(I:C) (20 μg/kg) every two days in the entire RT cycle (Fig. 7A). From December 2022 to June 2023, a total of 5 advanced HCC patients (P01, P02, P03, P04, P05) with local HCC lesions and distant metastatic tumor lesions were enrolled and received a complete 8 injections of poly(I:C) according to the trial procedure (Table EV1). Significantly, no obvious treatment-related adverse events were reported and routine blood/biochemical tests also showed no obvious abnormalities in those 5 enrolled patients during the whole trail period (Tables EV2 and EV3). Meanwhile, 5 control cases with advanced HCC that also treated with RT but without poly(I:C) injection during the same period were included. These control patients met our inclusion criteria and had at least one tumor lesion that were not be treated with RT (Table EV1). The irradiated tumor lesions and the non-irradiated targeting tumor lesions from all enrolled patients and control cases were evaluated according to the independent assessment with RECIST 1.1 within 1–2 months after RT treatment (Figs. 7B–D and EV5, Table EV4). For irradiated tumor lesions, there were 2 PR (tumor burden: −35.1% in P01; −31.2% in P03), 2 CR (tumor burden: −100% in P04 and P05), and 1 SD (tumor burden: −25.7% in P02) in enrolled 5 patients; and

2 PR (tumor burden: −54.7% in C04; −49.0% in C05), 1 CR (tumor burden: −100% in C03) and 2 SD (tumor burden: −7.3% in C01; −26.7% in C02) in the control cases. Significantly, for non-irradiated targeting tumor lesions, 3 PR (tumor burden: −49.0% in P01; −49.9% in P02; −63.5% in P03), 1CR (tumor burden: −100% in P04) and 1 SD (tumor burden: −23.3% in P05) were observed in enrolled 5 patients; while 4 SD (tumor burden: −25.9% in C01; 2.9% in C03; 8.5% in C04; 2.7% in C05) and 1 PD (tumor burden: 30.1% in C02) were found in the control cases. Although the number of enrolled patients currently is limited, they had relatively better efficacy in both irradiated tumor lesions (Enrolled patients: 2 PR/2 CR/1 SD Vs Control cases: 2 PR/1 CR/2 SD) and non-irradiated targeting tumor lesions (Enrolled patients: 3 PR/1 CR/1 SD Vs Control cases: 4 SD/1 PD) when compared with the control cases (Table EV4). Overall, these results suggested that poly(I:C) injection could enhance the anti-tumor efficacy of radiotherapy for advanced HCC patients, especially in terms of abscopal effect, which is consistent with the phenomenon found in HCC mouse models.

# Discussion

Clinically, RT could induce a series of RCD, accompanied by the release of tumor-relative antigens from irradiation-damaged tumor cells, which might act as an "in situ auto-vaccination" to trigger abscopal effect (Goto, 2019; Lei et al, 2021; Zhao and Shao, 2020). However, such phenomenon is a rare in clinic (Barazzuol et al, 2020; Mondini et al, 2020; Zhao and Shao, 2020). Presently, RT combined with immune checkpoint inhibitors (such as PD-1 antibody or CTLA-4 antibody) has been shown to induce about 29% rate of abscopal effect in patients with advanced tumors; nevertheless, the incidence of abscopal effect is still unsatisfactory, which may be caused by lack of tumor infiltrating activated T cells and further make it impossible to initiate an adequate immune response (Ozpiskin et al, 2019; Zhao and Shao, 2020). In addition, the side effects of immunotherapy was nonnegligible (Mandlik et al, 2023), which limits the combined application of radiotherapy and immunotherapy. Although the treatment regimen of RT combined with poly(I:C) has been proven to be feasible in patients with advanced HCC, the abscopal effect is not significant and the mechanism is unclear (de la Torre et al, 2017). Here, we optimized the combined treatment plan of RT and poly(I:C). For RT in HCC patients, we use therapeutic dose (3–5 Gy/fraction, a total dose of 40–50 Gy) to treat the targeted tumor lesion, which could release more tumor-associated antigens for immune system activation; secondly, in terms of the method and time of poly(I:C) injecting, subcutaneous injection of poly(I:C) was performed in every two days in the entire RT cycle, which could help to make full use of tumor-associated antigens released from RT treatment in real time

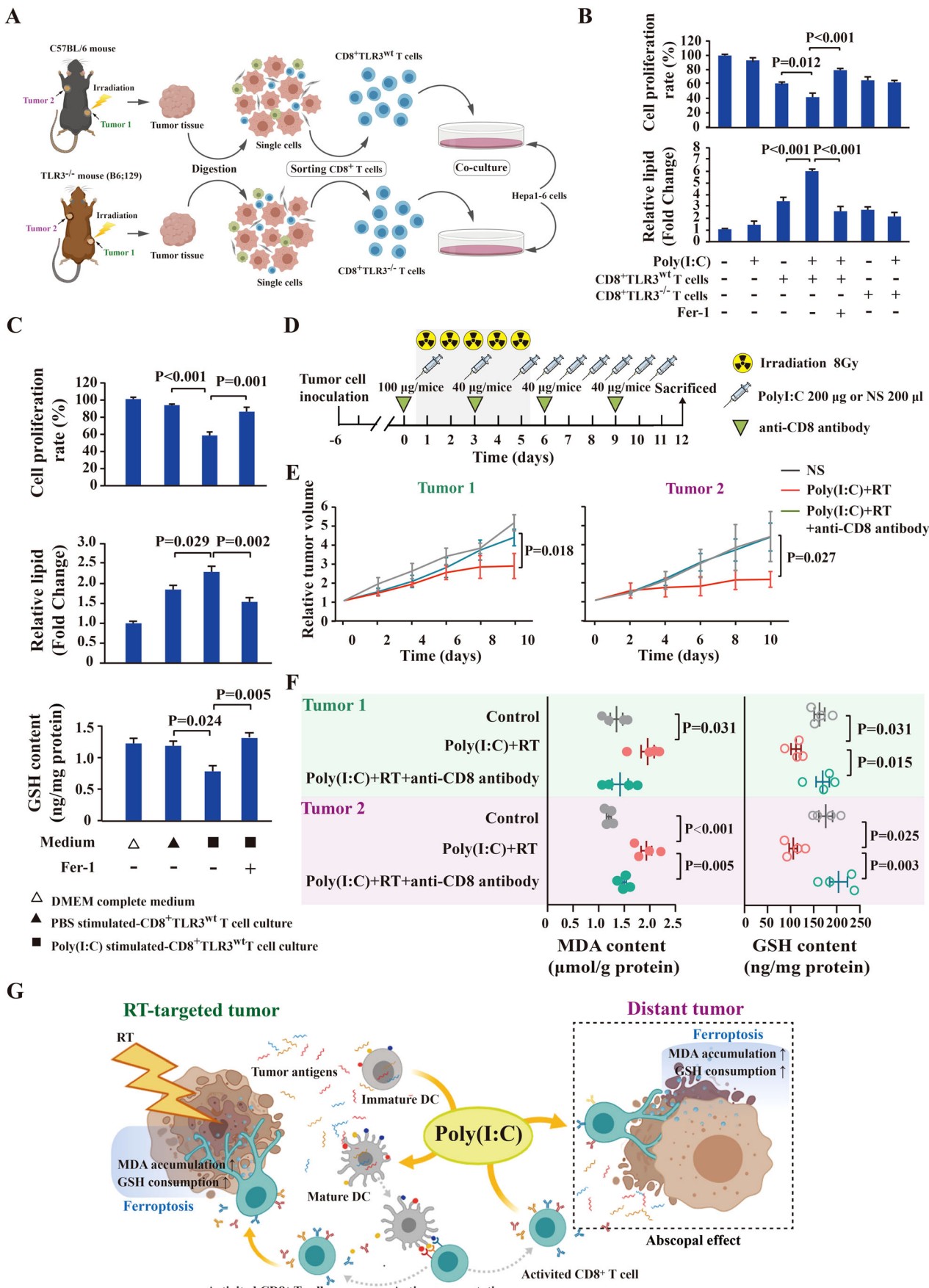

**Figure 6. The promoting effect of poly(I:C) on irradiation-induced tumor ferroptosis is dependent on the activation of TLR3 signaling of tissue-infiltrating CD8+ T cells.**

(A) Schematic diagram for Sorting of tumor-infiltrating CD8+ T cells derived from bilateral HCC subcutaneous tumor models constructed in C57BL/6 mouse or TLR3-knockout mouse and the co-culture with Hepa1-6 cells. (B) Cell proliferation rate (upper panel) and relative lipid (bottom panel) of Hepa1-6 cells with different intervention. Cell proliferation rate was assessed by CCK-8 assay. The cell proliferation rate and relative lipid of cells grown under normal conditions were normalized to 1. The data represent the means ± SEM (error bar) of at least three independent experiments, and the differences among groups were analyzed by ANOVA. (C) Cell proliferation rate (upper panel), relative lipid (middle panel), and GSH content (bottom panel) of Hepa1-6 cells grown under different medium analyzed by CCK-8 assay. The cell proliferation rate and relative lipid of cells grown under normal conditions was normalized to 1. The data represent the means ± SEM (error bar) of at least three independent experiments, and the differences among groups were analyzed by ANOVA. (D) Schematic diagram of anti-CD8 antibody treatment in mouse HCC tumor model receiving poly(I:C) plus RT. (E) Relative tumor growth curves of Tumor1 and Tumor2 in different treated group ($n = 4$ mice for each group) as indicated. Results are shown as mean ± SEM (error bar). (F) Determination of MDA content (left panel) and GSH content (right panel) in Tumor1 and Tumor2 in different treated group ($n = 5$ independent samples for each group) as indicated (illustrated by scatterplot). The dada are shown as mean ± SEM (error bar), and ANOVA was performed to analyze the differences among groups. (G) Schematic illustration of poly(I:C) regulating tumor ferroptosis to promote the abscopal effects of RT. Source data are available online for this figure.

to promote antigen presentation and CD8+ T cell activation. Meanwhile, because this treatment plan does not use invasive treatment measures, it is clinically operable and has a high degree of patient acceptance. In terms of therapeutic efficacy, through animal models and prospective clinical studies, we have found the same phenomenon, that is, when the HCC mice models or HCC patients are further subcutaneously injected with poly(I:C) during RT for the target lesion, the abscopal effect can be significantly enhanced.

Polyinosinic-polycytidylic acid [Poly(I:C)] injection for clinical use was successfully adopted in 1971. As a synthetic dsRNA, poly(I:C) could activate host defense in a pattern similar to that of viral infection, and regulate antigen-specific immune responses as immunomodulators through pathogen-associated molecular patterns by bounding to antigens properly, thereby stimulating cell-mediated immune responses to kill viruses (De Waele et al, 2021). Therefore, it was widely used as a clinical broad-spectrum antiviral drug in China with high safety and very low side effects for half a century, including the treatment of hepatitis B. However, with the rapid development of antiviral drugs, more powerful antiviral drugs significantly superior to poly(I:C) were emerging continuously, therefore the clinical application of poly(I:C) is greatly reduced. Recently, with the rapid development of tumor vaccines, poly(I:C) and its derivatives have been widely used as vaccine adjuvants in various tumor vaccine-related clinical trials around the world (Cai et al, 2021; Le Naour et al, 2020; Ogino et al, 2022), demonstrating a safe and efficient features for activating the body's immune response. In this study, antigens released from irradiated tumor cells were similar to in situ auto-vaccination, and poly(I:C) could enhance the effect of this vaccination through promoting the presentation of antigens released from irradiated tumor cells to recruit activated CD8+ T cells to the tumor sites. Although the enrollment of patients is limited in our study, RT combined with poly(I:C) has proven to be a safety and feasible strategy in HCC. Importantly, it is very cheap, with the price only one-hundredth of that of immune checkpoint inhibitors, making it very suitable for large-scale clinical promotion, especially in developing countries. However, we also found that RT combined with poly(I:C) also cannot maintain this phenomenon for a long time, and new lesions will still occur later. This may be mainly because the highly expressed immune checkpoint receptors such as PD-L1 in the immunosuppressive microenvironment of HCC could bind to the PD-1 receptor of T cells infiltrating into tumor tissue, thus leading to T cell disfunction or depletion and the subsequent occurrence of

new tumor lesions. Our preliminary experiments in HCC mouse models also showed that further combination of α-PD1 on the basis of poly(I:C) and RT treatment can further enhance and prolong the efficacy and duration of abscopal effect (Appendix Fig. S5), which can be served as a further potential improvement scheme, but the effect remains to be further explored, and the related toxicity still need to be further throughly investigated.

In addition, the molecular mechanism by which RT combined with poly(I:C) improves the abscopal effect previously remains totally unclear. In this study, we report for the first time that induction of ferroptosis in HCC by tumor infiltrated CD8+ T cells that atctivated by TLR3 signaling is the key mechanism for improving the abscopal effect of the combined treatment. Ferroptosis is a kind of programmed cell death induced by lipid peroxidation, which can be seen in RT, chemotherapy and tumor immunotherapy, and has great potential in cancer treatment. Consistent with published reports, we also observed the genetic and biochemical characteristics of ferroptosis in HCC cells of target lesion after RT, such as increasing lipid accumulation and enhancing lipid peroxidation. More interestingly, we found that when poly(I:C) was further combined with RT, ferroptosis in target lesions or distant lesions was significantly enhanced, and the corresponding tumor control ability was also significantly improved. This phenomenon revealed ferroptosis as a novel mechanism for the synergistic effect of immunotherapy and RT. Notably, Wang et.al had demonstrated that immunotherapy-activated CD8+ T cells could promote lipid peroxidation of tumor cells and enhance tumor ferroptosis sensitivity (Wang et al, 2019). This study suggested that poly(I:C) could act as a promoter of CD8+ T cell-regulated of ferroptosis. Poly(I:C) could promote the antigen presentation and maturation of DC cells through the TLR3 signaling pathway after RT, and then promote T cells activation, proliferation, and infiltration into the irradiated and non-irradiated distant tumor tissue for enhancing tumor control in the HCC mouse model, thereby regulating the ferroptosis of tumor. Therefore, poly(I:C) synergized with RT could generate effective antitumor immunity by inducing ferroptosis. It is worth mentioning that previous studies have confirmed TLR3 activation promoted NK cell activation and cytotoxicity in vitro (Chew et al, 2012), thus, it will also be interesting to investigate whether poly(I:C) also exerts anti-tumor effects by activating the TLR3 receptor of NK cells.

In conclusion, this study demonstrated for the first time that poly(I:C) could enhance the RT-induced ferroptosis of HCC cells by promoting the antigen presentation ability of DCs, as well as the tumor infiltration and killing ability of CD8+ T cells, thus increasing the frequency and potency of abscopal effect of RT in

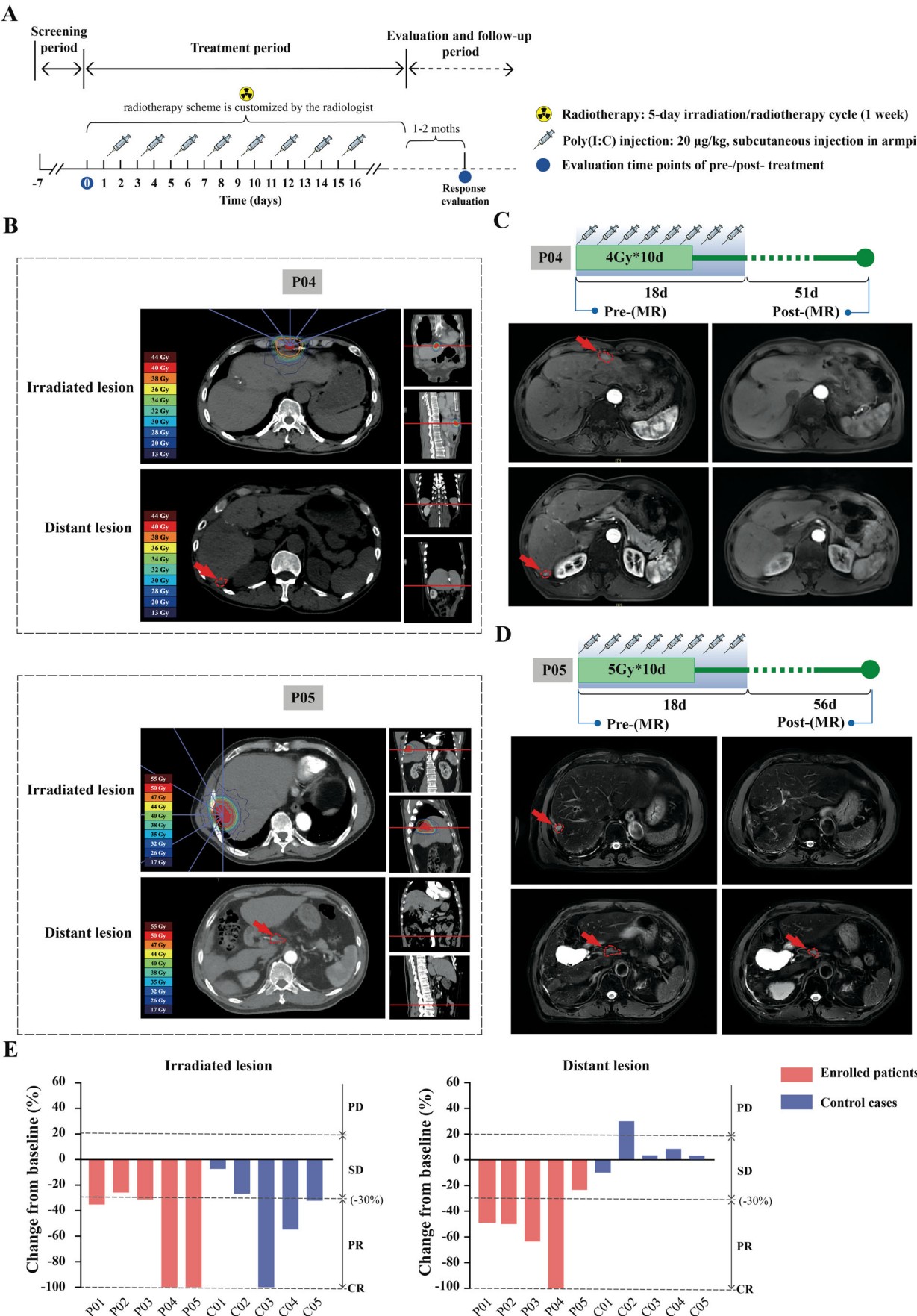

**Figure 7. Poly(I:C) improves abscopal effect of radiotherapy in patients with advanced HCC.**

(A) Treatment timeline of enrolled patients in the clinical trial. Clinical event timeline and corresponding represented imaging of all enrolled patients. (B) Dose distribution of radiotherapy of patient P04 (upper panel) and patient P05 (bottom panel). The radiotherapy schemes of patients were constructed on their CT imags. In the irradiated lesion, the indigo line indicates the irradiation path of each irradiation field, and the different colored closed-loop lines represent different irradiation dose (corresponding to the dose color block in the lower left corner of the panel). The red arrow marks the location of the distant lesion. The red line marks the horizontal position of the tumor lesion shown in the coronal and sagittal positions. (C) Representative MR scans of patient P04 (tumor lesions outlined in red and marked with red arrows): after radiotherapy, the irradiated HCC metastatic lesion located on peritoneum was eliminated, while the non-irradiated intrahepatic primary HCC lesion was also eliminated. (D) Representative MR scans of patient P05 (tumor lesions were outlined in red and/or marked with red arrows): the irradiated intrahepatic tumor was eliminated after radiotherapy, while the non-irradiated intraperitoneal lymph node metastasis was reduced by 23.3%. (E) Changes of tumor diameter length of enrolled patients and control cases between pre-treatment and post-treatment were determined according to RECIST1.1 criteria. The tumor burden from pre-treatment was served as the basline. Source data are available online for this figure.

HCC. Moreover, poly(I:C) combined with RT has also been confirmed to be a safe and feasible strategy for improving the abscopal effect of RT in patients with advanced HCC in clinical. Finally, due to the limited number of enrolled patients, the phenomenon of poly(I:C) combined with RT-induced ferroptosis to enhance the abscopal effect still needs to be further confirmed by large-scale randomized controlled clinical trials.

# Methods

## Cell culture

The murine HCC cell line Hepa1-6 cells (RRID:CVCL_0327) were purchased from American Type Culture Collection (ATCC, Manassas, USA) at 2018. The murine HCC cell line H22 cells (RRID:CVCL_H613) were purchased from IMMOCELL (Xiamen, China) at 2023. Hepa1-6 cells and H22 cells were maintained in Dulbecco's modified Eagle's medium (DMEM) (HyClone, Logan, USA) supplemented with 10% FBS (HyClone), 100 U/mL penicillin and 100 µg/mL streptomycin (Thermo Fisher Scientific, Waltham, USA) under a humid atmosphere at 37 °C with 5% $CO_2$. For mycoplasma detection, supernatants from cells cultured to suitable densities (Hepa1-6 cells confluence: 80%, H22 cells density: $10^6$/ml) were obtained and treated at 95 °C for 10 min. Subsequently, the processed samples were assessed according to the instructions of *TransDetecte*® PCR Mycoplasma Detection Kit (TransGen Biotech, Beijing, China). Both Hepa1-6 cells and H22 cells were assessed to be free of mycoplasma contamination. For cell authentication, cell lines were authenticated by multiplex PCR amplification of eighteen short tandem repeat (STR) loci. The authentication report of Hepa1-6 cell line was provided by Genetic Testing Biotechnology Co., Ltd. (Suzhou, China) on April 28, 2022, and authenticated as a 96.00% match to ExPASy cell line Hepa1-6. The last authentication report of H22 cell line was provided by IMMOCELL (Xiamen, China) on January 19, 2022, and authenticated as an exact match to H22 cell line included in ATCC, Deutsche Sammlungvon Mikroorganismenund Zellkulturen (DSMZ), Japanese Cancer Research Resources Bank (JCRB), and RIkagaku KENkyusho (RIKEN). Starting with the first culture in the laboratory, the passages of all cells used in all experiments were less than 20.

## In vivo tumor irradiation experiments

Male C57BL/6 mice (6–8 weeks old; RRID:MGI:2159769) and male BALB/c mice (6–8 weeks old; RRID:MGI:2161072) were purchased from China Wushi, Inc. (Shanghai, China), and allowed to acclimate in the institutional animal facility. 6–8-week-old TLR3-knockout mice (Homozygous TLR3$^{-/-}$, identifier: CSTR 16397.09.0G01000332; RRID:IMSR_JAX:005217) generated on a C57BL/6 background (B6;129S1-Tlr3tm1Flv/J) were purchased from National Human Disease Animal Model Resource Center (Beijing, China). The mice were housed in animal facility with controlled temperature (20–24 °C) and humidity (50–60%) on a 12-h light/dark cycle. They had ad libitum access to food and filtered water, and were daily monitored for health status by trained personnel. All animal experiments were conducted in strict accordance with the procedures approved by the Animal Ethics Committee of Mengchao Hepatobiliary Hospital of Fujian Medical University (Approval number: MCHH-AEC-2022-04). Hepa1-6 cells were subcutaneously injected into the right hind limbs (1.0 × $10^6$ cells in 0.1 ml/site, defined as Tumor1) and left forelimb armpit (5 × $10^5$ cells in 0.1 ml/site, defined as Tumor2) of C57BL/6 mice (or TLR3-knockout mice) on day 0, respectively. H22 cells were subcutaneously injected into the right hind limbs (2 × $10^6$ cells in 0.1 ml/site, defined as Tumor1) and left forelimb armpit (1 × $10^6$ cells in 0.1 ml/site, defined as Tumor2) of BALB/c mice on day 0, respectively. Seven days after the inoculation, the tumors were established (Tumor1 and Tumor2 were ~20 mm³ and ~25 mm³, respectively). The mice were randomly grouped and then started to receive corresponding treatment. For evaluating the efficacy of combing poly(I:C) with RT, the bilateral HCC subcutaneous tumor-bearing mice were randomly divided into four groups (n = 10 per group, the experiment was independently repeated twice, each time with 5 mice per group), including normal saline (NS) treated group, poly(I:C) treated group, RT treated group, and poly(I:C) plus RT treated group. The mice were administered with poly(I:C) (200 µg per mouse) (Nanguo, Zhanjiang, China) or NS (200 µl per mouse) by subcutaneous injected at bilateral groin area circularly, at the day1, day3, and day5–day11 of the treatment procedure. In the experimental group that received RT, the mice were anesthetized with intra-peritoneal injection of 50 mg/kg pentobarbital sodium before radiation. The right hind limb and tail of the mice were fixed on the plate to allow the right hind limb with the tumor (Tumor1) stretched out and exposed to the set irradiation field. All body parts except the right hind limb were protected from irradiation in the field. The Tumor1 in tumor-bearing mice was irradiated with a total dose of 40 Gy (8 Gy/fraction) using the Versa-HD linear accelerator (Elekta, Stockholm, Sweden) with the following irradiation parameters: voltage, 6 MV; direction, 0°; dose rate, 6 Gy/min; irradiated volume; distance from source to tumor center, 100 cm. To confirm the regulation of poly(I:C) on tumor ferroptosis, ferroptosis inhibitor ferrostatin-1 (Fer-1) (GlpBio, Montclair, USA) was administered by tail vein

injection at a dose of 0.8 mg/kg (diluted in normal saline, 200 μl per mouse) at the day6, day9, day12 of the treatment procedure, on the basis of poly(I:C) plus RT or RT alone. To eliminate the effect of ROS generated by RT, the ROS inhibitor acetylcysteine (N-acetylcysteine, NAC) (Selleck, Shanghai, China) was injected subcutaneously into HCC mice at dose of 100 mg/kg (diluted in normal saline, 100 μl per mouse) at the day 1–6 of the treatment procedure. To demonstrate the necessity of CD8[+] T cells in poly(I:C)-mediated tumor ferroptosis, poly(I:C) plus RT-treated mice were intraperitoneally injected with 100 μg of anti-CD8 antibody (500 μg/ml; Leinco, St. Louis, USA; RRID:AB_2737483) 1 day before RT, and followed by periodic depletion of CD8[+] T cells through injection of 40 μg of anti-CD8 antibody (400 μg/ml) every 3 days until the end of the experiment. The tumor volumes were measured every 2 days. Once the maximum diameter of the tumor exceeded 15 mm or tumor ulceration occurred, the mice were sacrificed. Relative tumor volume is calculated as $V_t/V_0$. $V_0$ is the tumor volume before treatment, $V_t$ is each measurement of the corresponding tumor volume. Mice in the efficacy evaluation experiment and mice in the in vivo rescue experiment were sacrificed at the day20 and at least the day12 after the initial treatment, respectively. Fresh tumor tissues were harvested and divided into several tumor specimens for downstream experiments. DCs and splenic monocytes were freshly obtained from lymph nodes and spleens, respectively.

## Bulk RNA-seq and data analysis

Total RNA of C57BL/6 mouse tumor tissues was extracted by using EasyPure RNA kit (Transgen Biotech, Beijing, China) according to the manufacturer's instructions. Then RNA samples were subjected to RNA library preparation using VAHTS Stranded mRNA-seq Library Prep Kit. RNA library sequencing was performed by Berry Genomics (Bejing, China) on Illumina Novoseq 6000 (paired end, 150 bp). The raw sequencing reads of RNA-seq data were first preprocessed with fastp (v0.21.0) to remove unqualified reads, and remaining reads were aligned to mm10 mouse genome using STAR (v2.7.8a, RRID:SCR_004463). To quantify the expression levels for each gene, TPM (transcripts per kilobase of exon model per million mapped reads) was calculated by RSEM (v1.3.0). Enrichment scores of various RCD (Galluzzi et al, 2018; Karki and Kanneganti, 2019; Liu et al, 2020; Miao et al, 2022; Miao et al, 2020; Tang et al, 2019; Wang and Liu, 2021; Xia et al, 2019; Zhou and Bao, 2020) were estimated using single sample gene set enrichment analysis (GSEA, ssGSEA, RRID:SCR_005724) by the gene set variation analysis ('GSVA') R package (version 1.42.0). The hallmark gene sets were obtained from MSigDB database from GSEA software (https://www.gsea-msigdb.org/gsea/downloads.jsp). The gene lists of various RCD were showed in Dataset EV1. The infiltration scores of multiple immune cells were estimated by the online analysis site ImmuCellAI (http://bioinfo.life.hust.edu.cn/ImmuCellAI#!/ImmuCellAI), which is a tool to estimate the abundance of 36 immune cells, including 16 T cells (CD8[+] T cells, CD4[+] T cells, Treg, etc.), 2 Macrophages (M1, M2), B cell, NK cell, Monocytes, Neutrophils, etc. The ssGSEA was performed to assess the relative level of immune cell infiltration, and the enrichment score of each immune cell type was used to represent the relative level of tumor immune infiltration score (Miao et al, 2022; Miao et al, 2020). Gene Ontology (GO) annotation and Kyoto Encyclopedia of Genes and Genomes (KEGG) pathway enrichment

analysis were performed through an analysis module of Bioinformatics (www.bioinformatics.com.cn) that is integrated with clusterProfiler and pathview (Yu et al, 2012). $P < 0.05$ were considered significantly enriched. The $-\log10$ ($p$-value) denotes enrichment score showing the significance of the GO function and pathway correlations.

## qRT-PCR

Total RNA was isolated from fresh-frozen mouse tumor tissues and vascular cells sorted from tumor tissues using TransZol reagents (TransGen Biotech), respectively. Reverse transcription was carried out using a Transcriptor First Strand cDNA Synthesis Kit (Roche, Indianapolis, USA) according to the manufacturer's instructions. The primers for *Vcam-1*, *CXCL9*, *CXCL10*, *CCL2*, *CCL3*, *CCL4*, *CCL5*, *CCL8*, and *CCL11* were designed to determine the abundance of corresponding mRNA. The primers are listed in Table EV5. For qRT-PCR analysis, aliquots of double-stranded cDNA were amplified using SYBR Premix Ex Taq II (Takara, Dalian, China) in the StepOne™ Real-Time PCR System. The amplification parameters were as follows: pre-denaturation at 95 °C for 10 min, denaturation at 95 °C for 30 s, annealing at 60 °C for 30 s, extension at 72 °C for 25 s, a total of 40 cycles. Relative RNA expression was calculated with $2^{-\Delta\Delta Ct}$ method, and then normalized to the geometric mean of the GAPDH. qRT-PCR was performed at least three times for each RNA.

## Lipid peroxidation and ROS detection

About 25 mg fresh tumor tissue was homogenized with lysis buffer [2% SDS (Sigma), and 1× protease inhibitor (Roche) in PBS] to obtain tumor tissue lysate. The level of malondialdehyde (MDA) in tumor tissue lysates was evaluated according to the instructions of Lipid Peroxidation Assay Kit (Cayman, Ann Arbor, USA). Then, the absorbance at 532 nm was measured using spectrophotometer. Intracellular ROS of tumor tissue was assessed according to the instructions of Reactive Oxygen Species (ROS) Fluorometric Assay Kit (Elabscience, Wuhan, China). In brief, tumor tissues were digested into single cells and incubated with DCFH-DA for 1 h at 37 °C in the dark, and washed three times with PBS subsequently. Then, the fluorescence intensities reflecting the intracellular ROS were detected by spectrophotometer at 525 nm absorbance. For lipid peroxidation study, the cells were incubated with 10 μM sensor probe BODIPY(581⁄591)-C11 (Thermo Fisher Scientific) for 1 h at 37 °C. Then, Excess BODIPY-C11 probe was removed by washing the cells twice with PBS. Labeled cells were trypsinized, and then re-suspend in PBS containing 5% BSA. The fluorescence intensity of each sample was determined by flow cytometry (BD FACSVerse™, BD, USA). The shifted fluorescence of emission peak from 590 to 510 nm in each sample was analyzed and used to calculate the relative lipids.

## Glutathione content assay

About 25 mg fresh tumor tissue was homogenized with lysis buffer (as described above) to obtain tumor tissue lysate. Cells were washed and scraped off with cell scraper and then suspended in lysis buffer (as described above) to obtain cell lysate. Reduced glutathione (GSH) content in tumor tissue lysates and cell lysates were measured using Reduced Glutathione (GSH) Colorimetric

Assay Kit (Elabscience, Wuhan, China) and the absorbance was measured with microplate reader at 405 nm according to the manufacturer's instructions.

## Tissue processing for flow cytometry

Lymphocytes in lymph node and splenic monocytes were isolated as described in previous studies (Chen et al, 2022). In brief, to isolate lymphocytes in lymph node, the mouse lymph nodes were mashed on a 40 μm cell strainer, and then the obtained single-cell suspension was centrifuged at $800 \times g$ for 5 min to obtain lymph node lymphocytes. To isolate splenic monocytes, spleens were excised and smashed with syringe plunger to obtain spleen cell suspension, and then splenic monocytes were isolated by Ficoll density gradient centrifugation from spleen cell suspension.

To obtain single cells (including vascular cells) in tumor tissue, $3–4 \ mm^3$ of tumor specimens were minced and digested as described in previous studies (Chen et al, 2022). In brief, minced tumor specimens were digested by gentle stirring in 37 °C in RPMI-1640 medium containing 1 mg/mL collagenase type II (Sigma, Darmstadt, Germany) and 2 mg/mL DNase I (Sigma). The single-cell suspension from tissue digestion was washed and filtered through a 40 μm cell strainer to obtain single-cell suspension. The monocytes were isolated from single-cell suspension by the Ficoll density gradient centrifugation.

## Flow cytometric sorting

Single-cell suspensions of tumor tissue (as described above) and splenic monocyte suspensions (as described above) were blocked with PBS containing 5% BSA for 20 min in room temperature. Subsequently, tissue single-cells were stained with anti-mouse CD45-APC mAb (1:200 dilution; eBioscience™, RRID: AB_469392), anti-mouse CD8-FITC mAb (1:100 dilution; eBioscience™, RRID: AB_464915) and anti-mouse CD31-PE mAb (1:100 dilution; eBioscience™, RRID: AB_465632); Splenic monocytes were stained with anti-mouse CD45-APC mAb (1:200 dilution; eBioscience™, RRID: AB_469392), anti-mouse CD3-FITC mAb (1:100 dilution; eBioscience™, RRID: AB_2572431). After 30 min of staining, the cells were washed twice with PBS and re-suspended in 0.5% BSA (in PBS). Sorting was performed on a Fusion cell sorter (BD FACSAria™, USA). In the FSC/SSC-gated monocytes, the $CD45^+CD8^+$ single-cells of tumor tissue were recognized as tumor-infiltrating $CD8^+$ T cells, the $CD45^-CD31^+$ single-cells of tumor tissue were recognized as vascular endothelial cells, the $CD45^+CD3^+$ splenic monocytes were recognized as splenic T cells.

## Flow cytometry

Single-cell suspensions from tumor and monocytes derived from lymph nodes and spleen were blocked in 5%BSA (in PBS) or Fcblock for 20 min. For flow cytometry analysis of DC maturity, the monocytes derived from lymph nodes were stained in 0.5% BSA for 30 min with anti-mouse CD11c-PE mAb (1:100 dilution; eBioscience™, Waltham, USA; RRID: AB_465552), anti-mouse CD80-APC mAb (1:200 dilution; eBioscience™; RRID: AB_469417), anti-mouse CD86-PE/Cy7 mAb (1:100 dilution; eBioscience™; RRID: AB_2573372). For flow cytometry analysis of memory T cells in the spleen, monocytes derived from spleen were

stained with anti-mouse CD45-APC mAb (1:200 dilution; eBioscience™; RRID: AB_469392), anti-mouse CD8-FITC mAb (1:100 dilution; eBioscience™; RRID: AB_464915), anti-mouse CD62L-PerCP/Cy5.5 mAb (1:100 dilution; eBioscience™; RRID: AB_996667), and anti-mouse CD44-PE/Cy7 mAb (1:100 dilution; eBioscience™; RRID: AB_469623). The above fluorescent antibody staining is incubated in 0.5% BSA for 30 min, rotated in the dark. Subsequently, the cells were washed twice with PBS and then re-suspended in PBS with 0.5% BSA. For Annexin V/PI staining, $0.5–1 \times 10^6$ tumor cells were incubated with Annexin V and propidium iodide (PI) for 15 min in dark, using Annexin V-FITC/ PI Cell Apoptosis Detection Kit (TransGen Biotech, Beijing, China) according to the manufacturer's instructions. Flow cytometry analysis was performed on BD FACSVerse™ and FlowJo v.10.

## Enzyme-linked immunospot (ELISpot) assay

IFN-γ secretion of mouse splenic T cells were detected by ELISpot kit (Mabtech, Cincinnati, US). The bone marrow derived-DCs (BMDCs) were isolated from femurs and tibias of 6–8-week-old naive C57BL/6 mice. On the sixth day of in vitro culture, BMDCs ($3 \times 10^4$/well) were seeded into duplicate wells of ELISpot plates and pulsed with 4 μg neoantigen peptides (Genscipt, Nanjing, China) for 24 h. Subsequently, neoantigen peptides-pulsed BMDCs were co-cultured with spleen T cells ($3 \times 10^5$/well) for 48 h. The plates were washed before the addition of the detection antibody (R4-6A2-biotin, 1:100 dilution) and the incubated with primary antibody of IFN-γ for 1 h in room temperature. Streptavidin-ALP (1:1000 dilution) was subsequently added and incubated at room temperature for another 1 h. TMB substrate solution was then added to each well. After ~5 min of incubation, the reaction was stopped by adding deionized water. Finally, the AT-Spot-2200 ELISPOT Analysis System (Antai Yongxin, Beijing, China) was used to scan the plates and analyze the IFN-γ spot-forming cells.

## Tissue cytokine measurement

Tumors tissues were isolated on ice, weighed, and homogenized in PBS containing 5 μl protease inhibitor Cocktail (Roche) per mg tissue. Supernatants of tissue homogenate were harvested following centrifugation at $12,000 \times g$ for 20 min at 4 °C. The level of mouse TNF-α, IFN-γ, GzmB, IL-2, IL-6, IL-12 was assessed using mouse ELISA kits (Boster, Wuhan, China) following the manufacturer's instructions.

## Immunofluorescence

Isolated mice tumor tissues were formalin-fixed and paraffin-embedded. Then, sections of 4-μm thickness tissue were prepared on slides. For immunofluorescence with single marker, tissue slides of mouse tumor were blocked with PBS containing 5% (v/v) BSA for 20 min at room temperature, and then incubated at 4 °C overnight with primary antibodies [anti-mouse CD8 mAb (1:500 dilution; Servicebio, Wuhan, China), anti-mouse Granzyme B (GzmB) mAb (1:50 dilution; CST, Danvers, USA; RRID: AB_2799313)] followed by incubation with secondary antibodies (Servicebio) for 30 min at room temperature. Cell nuclei were counterstained with DAPI (Invitrogen, Waltham, USA). Images of immune-stained slides were collected by fluorescent microscope

(NIKON ECLIPSE C1). Multi-color immunofluorescence is performed according to Opal 6-Plex Detection Kit (Akoya Biosciences, Menlo Park, USA; RRID: AB_2814585). Slides were stained with anti-mouse CD8α mAb (1:500 dilution; Abcam, Cambridge, UK; RRID: AB_2890649), anti-mouse TLR3 mAb (1:200 dilution; Bioss, Hong Kong, China; RRID: AB_10855240), anti-mouse Granzyme B (GzmB) mAb (1:200 dilution; CST, Danvers, USA; RRID: AB_2799313), anti-mouse CD45 mAb (1:50 dilution; Abcam, Cambridge, UK; RRID: AB_442810), anti-mouse CD11c mAb (1:200 dilution; CST, Danvers, USA; RRID: AB_2800282), anti-mouse CLEC9 mAb (1:100 dilution; Bioss, Hong Kong, China), anti-rabbit IL12 mAb (1:200 dilution; Solarbio, Beijing, China). Cell nuclei were counterstained with DAPI. Images of immune-stained slides were collected and analyzed by PhenoCycler-Fusion 2.0 (Akoya Bioscience).

## CCK-8 assay

Irradiated or non-irradiated Hepa1-6 cells were seeded in 96-well plates at a density of $3 \times 10^4$ cells per well and incubated at 37 °C in 5% $CO_2$ incubator. After the corresponding intervention, an aliquot of 10% (v/v) CCK-8 reagent (TransGen Biotech) was added to each well and incubated at 37 °C for 2 h. The absorbance (A) was measured at the wavelength of 450 nm according to the manufacturer's protocol. Samples were then measured every 24 h after 4 days of culture. The experiments were repeated independently for three times.

## In vitro stimulation and co-culture of tumor-infiltrating CD8$^+$ T cells

To mimic CD8$^+$ T cell recruitment in distant tumors in vivo, a co-culture system of CD8$^+$ T cells and Hepa1-6 cells was established. In detail, Hepa1-6 cells ($6 \times 10^4$/well) were seeded in 96-well plates and cultured at 37 °C in 5% $CO_2$ for 24 h. Subsequently, the tumor-infiltrating CD8$^+$ T cells (sorted by flow cytometry as described above) were added to the corresponding wells at $3 \times 10^5$ per well to co-culture with Hepa1-6 cells. Meanwhile, 10 μg/ml poly(I:C) was added to the corresponding wells once a day for two consecutive days.

To investigate the direct effect of CD8$^+$ T cell secreted proteins in supernatant on Hepa1-6 cells, CD8$^+$ T cell supernatant was directly co-cultured with Hepa1-6 cells. In detail, the sorted tumor tissue infiltrating-CD8$^+$ T cells were cultured in T cell medium (RPMI-1640 medium with 20 U/ml IL-2 and 5%FBS) with or without poly(I:C) (10 μg/ml) for 24 h, and then the supernatant was obtained and incubated with Hepa1-6 cells for 48 h.

## Quantification of cell culture supernatant protein by mass spectrometry

The supernatant of CD8$^+$ T cells with or without poly(I:C) stimulation described above were precipitated with ice-cold acetone (Sigma) and centrifuged at $10,000 \times g$ for 15 min at 4 °C. The precipitate was subsequently recalibrated with 200 μL lysis buffer [8 M urea (Sigma), 2% SDS (Sigma), and 1× protease inhibitor (Roche) in PBS] and incubated at 55 °C for 1 h. 10 μL iodoaceta-mide (500 mM, Sigma) was added to each sample and incubated for 30 min at room temperature under dark. The samples of protein

mixture were then ultrafiltered (molecular weight cut off 10,000 Da) and dissolved in 100 mM triethyl ammonium bicarbonate (Sigma). The protein digestion and iTRAQ labeling followed our previous reported procedures with modification (Liu et al, 2018; Xing et al, 2019). Briefly, proteins were digested with trypsin, and the peptide mixture was further labeled using the iTRAQ reagent kit (AB SCIEX, Framingham, USA). Equal amount of 8 labeled samples were combined and cleaned up by Sep-Pak Vac C18 cartridges (Waters) and then dried in a vacuum centrifuge for 2 h at 4 °C.

The LC-MS/MS analysis follows our previously reported protocol with certain modification. Briefly, 4 μg peptides were, respectively, re-suspended with 16 μl solvent A (0.1% formic acid in water), and separated by an EASY-nLC1000 system (Thermo Fisher Scientific), and then analyzed by a quadrupole-Orbitrap mass spectrometer (Q-Exactive Plus) (Thermo Fisher Scientific) equipped with an online nano-electrospray ion source. MS data acquiring was processed using Proteome Discoverer (Thermo Fisher Scientific; version 1.4, RRID:SCR_014477) against the uniprot human_database (released at 2020-06-18, 20,369 entries). Proteins with a fold change ≥2 ($P < 0.05$) in the supernatant with and without poly(I:C) stimulation were considered as differentially expressed secretory protein (DESP).

## Clinical trial design and intervention

This trial was an investigator initiated, single-arm, open-label clinical trial at Mengchao Hepatobiliary Hospital of Fujian Medical University in China and was registered at Chinese Clinical Trial Registry (http://www.chictr.org.cn/; ChiCTR2100053441). The main aim of this trial is to study the safety and feasibility of poly(I:C) in improving the abscopal effect of RT in patients with advanced HCC. The primary end point was the safety of poly(I:C) in combination with RT according to CTCAE 4.03 criteria. The secondary end point was the objective response rate (ORR) according to RECIST1.1 criteria. This clinical trial was approved by the Institution Review Board of Mengchao Hepatobiliary Hospital of Fujian Medical University (Fujian, China; Approval number: KESHEN 2021_135_02). All experiments in this trial conformed to the principles set out in the WMA Declaration of Helsinki and the Department of Health and Human Services Belmont Report, as well as the International Conference on Harmonization Guidelines for Good Clinical Practice. The informed written consents were signed by all enrolled patients. The key inclusion criteria of enrolled patients is as follows: (1) 18 to 75 years old male and female, with platelet count not lower than $50 \times 10^9$/L and creatinine clearance rate not lower than 60 ml/min; (2) CNLC stage IIa-IIb patients who are not suitable/unwilling to undergo surgery, ablation or interventional therapy, and CNLC stage IIIa-IIIb patients; (3) According to RECIST v1.1, there is at least one targeted tumor lesion (irradiated tumor lesions and/or non-irradiated targeting tumor lesions) in the liver that could be measrured by MR/CT imaging. The response evaluation standard [PD (progressive disease), SD (stable disease), PR (partial response) and CR (Complete response)] for irradiated tumor lesions and non-irradiated targeting tumor lesions were evalutated by two experienced physicians according to RECIST v1.1, respectively. Key exclusion criteria included: (1) Patients who received immunotherapy such as anti-PD-1/anti-PD-L1, or TACE within 6 months before enrollment; (2) Patients with HIV infection, HCV infection, serious coronary artery disease or other diseases that the researchers consider not suitable to be included in this study; (3) Patients with history of bone marrow transplantation or organ transplantation; (4) Patients with any

**The paper explained**

**Problem**

Beyond eliciting direct damage to irradiated tumor cells, radiotherapy (RT) also stimulates the anti-tumor immune response towards non-irradiated distant tumors. This phenomenon, termed the "abscopal effect", could hold significant promise for improving patients' quality of life and extending their survival. Nevertheless, RT-induced abscopal effect in advanced hepatocellular carcinoma (HCC) patients is rare, and effective clinical strategies to augment RT-related abscopal effect are lacking.

**Results**

We developed a novel combined treatment comprising RT and administration of the Toll-like receptor 3 (TLR3) agonist poly(I:C) in a noninvasive manner to enhance the abscopal effect in HCC. Administering poly(I:C) subcutaneously during RT treatment enhanced the abscopal effect in bilateral subcutaneous HCC mouse models, which was abolished by TLR3 knock-out or ferroptosis inhibitor. Mechanically, poly(I:C) improved ferroptosis in RT-targeted tumor and distant non-targeted tumor by promoting tumor antigen presentation by the dendritic cells and by stimulating activated CD8$^+$ T cell infiltration to secrete ferroptosis-related factors. Finally, the safety and feasibility of poly(I:C) combined with RT to enhance the abscopal effect were further confirmed through prospective clinical trial in advanced HCC patients.

**Impact**

This study presents a safe and effective clinical strategy to enhance the abscopal effects in advanced HCC with relative low cost and high feasibility, and provides new perspective to improve the therapeutic effects of RT.

form of immunodeficiency or history of autoimmune disease. Patient's radiotherapy scheme (3–5 Gy/fraction, a total dose of 40–50 Gy) is a tailor-made customization based on the patient's tumor size and location by the radiologist. Moreover, all enrolled patients druing RT treatment received poly(I:C) injections (20 µg/kg, subcutaneously in the left and right axillary lymphatic area, circularly. All patients who completed the trial with at least one efficacy evaluation were included in this study, and matched control cases were also included during the same period according to 1:1 propensity score matching (PSM).

## Statistical analysis

Statistical analysis was performed using SPSS software (version 20.0; IBM Corp.; RRID:SCR_002865) and R software (version 3.4.3; R Foundation for Statistical Computing). The t-test was applied to analyze the statistical significance between two groups of normally distributed dataset. The Mann-Whitney U test was applied to analyze the statistical significance between two datasets without normal distribution. One-way variance (ANOVA) was used to analyze the statistical significance of differences between more than two groups, followed by Bonferroni correction for multiple comparisons. The differences in distant tumor progression of mouse HCC models of different intervention groups were estimated using the Kaplan–Meier method, and the log-rank test was performed to examine the significance. The linear dependence was analyzed by Spearman's correlation. With the small sample size of RNA sequencing ($n = 3$ samples for each group), $P < 0.10$ was considered as statistically significant for corresponding analysis. Meanwhile, $P < 0.05$ was used as

the threshold of statistically significances in the analysis of other experimental data.

## Data availability

The transcriptome sequencing data and mass spectrometry data from this publication have been deposited to the Genome Sequencing Achieve (GSA) database (URL: https://ngdc.cncb.ac.cn/gsa/browse/CRA009656) and Open Archive for Miscellaneous Data (OMIX) database (URL: https://ngdc.cncb.ac.cn/omix/release/OMIX006088) of China National Center for Bioinformation, respectively, both with identical assigned identifier PRJCA014521. The source data of this paper are collected in the following database record: biostudies:S-SCDT-10_1038-S44321-024-00068-4.

## Peer review information

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

## Acknowledgements

This work was supported by the Major Research Projects for Young and Middle-aged Talent of Fujian Provincial Health Commission (2022ZQNZD014), Scientific Foundation of Fujian Province (Grant No. 2023J06049 and 2022J011290), Fuzhou Science and technology planning project (Grant No. 2021-S-112), Fuzhou Science and technology planning project (Grant No. 2023-R-003 and 2022-Y-006), the Science and Technology Plan Project of Fuzhou Health and Health System (Grant No. 2022-S-wt2), Fuzhou "14th Five-Year" clinical key specialty (Grant No. 20220203).

## Author contributions

**Liman Qiu**: Conceptualization; Data curation; Formal analysis; Validation; Investigation; Methodology; Writing—original draft; Project administration; Writing—review and editing. **Hongbing Ji**: Data curation; Formal analysis; Funding acquisition; Project administration. **Kai Wang**: Validation; Investigation; Methodology. **Wenhan Liu**: Methodology. **Qizhen Huang**: Conceptualization; Methodology. **Xinting Pan**: Formal analysis. **Honghao Ye**: Methodology. **Zhenli Li**: Methodology. **Geng Chen**: Methodology. **Xiaohua Xing**: Formal analysis; Methodology. **Xiuqing Dong**: Methodology. **Ruijing Tang**: Conceptualization; Methodology. **Haipo Xu**: Methodology. **Jingfeng Liu**: Conceptualization; Methodology. **Zhixiong Cai**: Conceptualization; Supervision; Validation; Investigation; Funding acquisition; Writing—review and editing.

**Xiaolong Liu**: Conceptualization; Supervision; Funding acquisition; Project administration; Writing—review and editing.

Source data underlying figure panels in this paper may have individual authorship assigned. Where available, figure panel/source data authorship is listed in the following database record: biostudies:S-SCDT-10_1038-S44321-024-00068-4.

## Disclosure and competing interests statement

The authors declare no competing interests.

# Expanded View Figures

**Figure EV1.   Effects of Fer-1 and NAC on tumor ferroptosis and tumor control induced by poly(I:C) plus RT in HCC mice model.**

(A) Tumor growth curves of directly irradiated tumor (Tumor1) and distant tumor (Tumor2) in different treated groups ($n = 5$ mice for each group) as indicated. Results are shown as mean ± SEM (error bar), and ANOVA was performed to analyze the differences among groups. (B) Schematic diagram of ROS inhibitor NAC treatment in HCC tumor mouse model receiving poly(I:C) plus RT. (C) Relative ROS level of Tumor1 and Tumor2 (obtained from mice sacrificed on day7 of the treatment procedure) in different treated groups ($n = 5$ independent samples for each group) as indicated (illustrated by scatterplot). The data are shown as mean ± SEM (error bar), and ANOVA was performed to analyze the differences among groups, and ANOVA was performed to analyze the differences among groups. (D) Tumor growth curves of directly irradiated tumor (Tumor1) and distant tumor (Tumor2) in different treated groups ($n = 5$ mice for each group) as indicated. Results are shown as mean ± SEM (error bar), and ANOVA was performed to analyze the differences among groups. Determination of MDA content (E) and GSH content (F) in Tumor1 and Tumor2 collected from different treated groups ($n = 5$ independent samples for each group) as indicated (illustrated by scatterplot). The data represent the mean ± SEM (error bar), and ANOVA was performed to analyze the differences among groups.

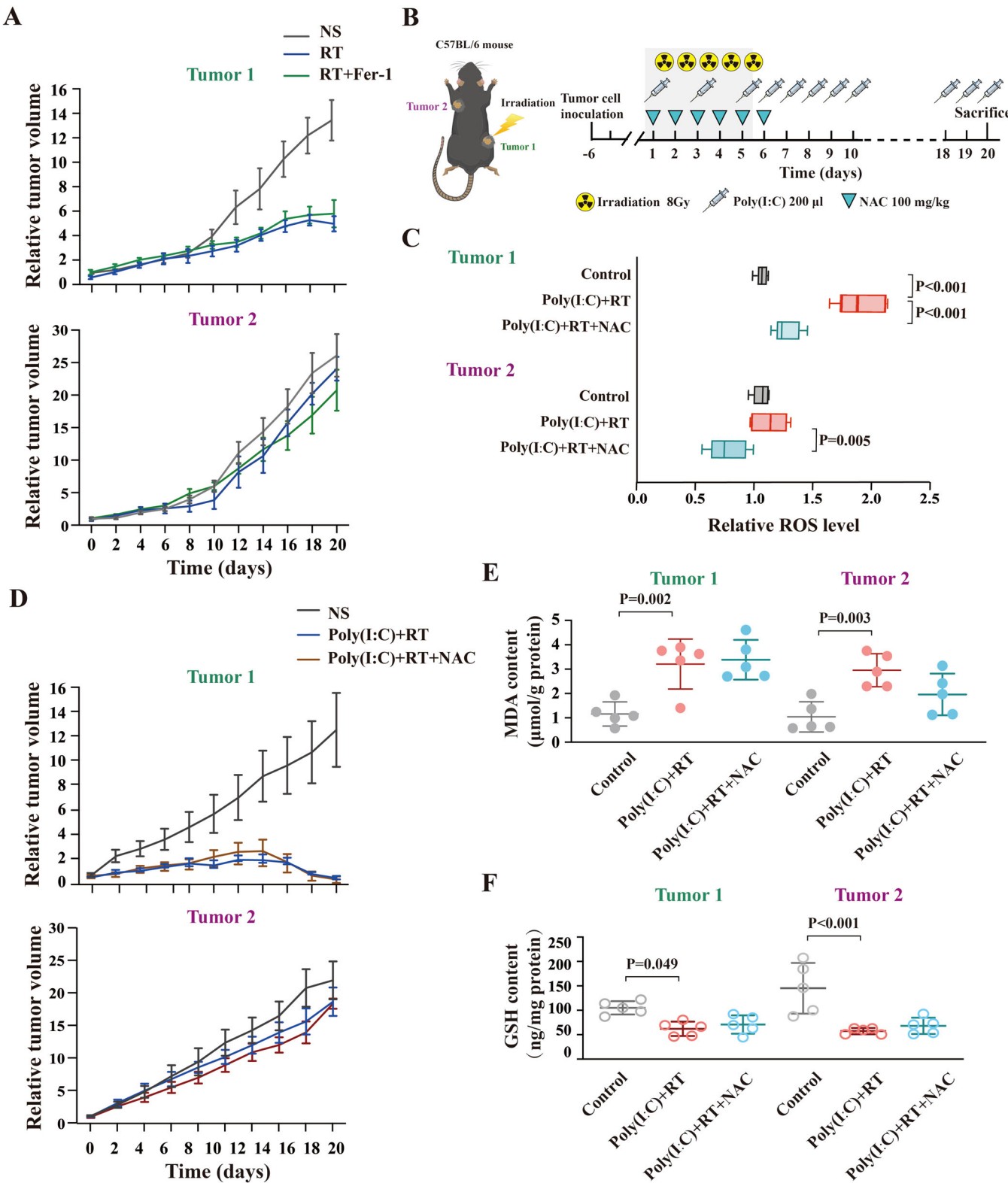

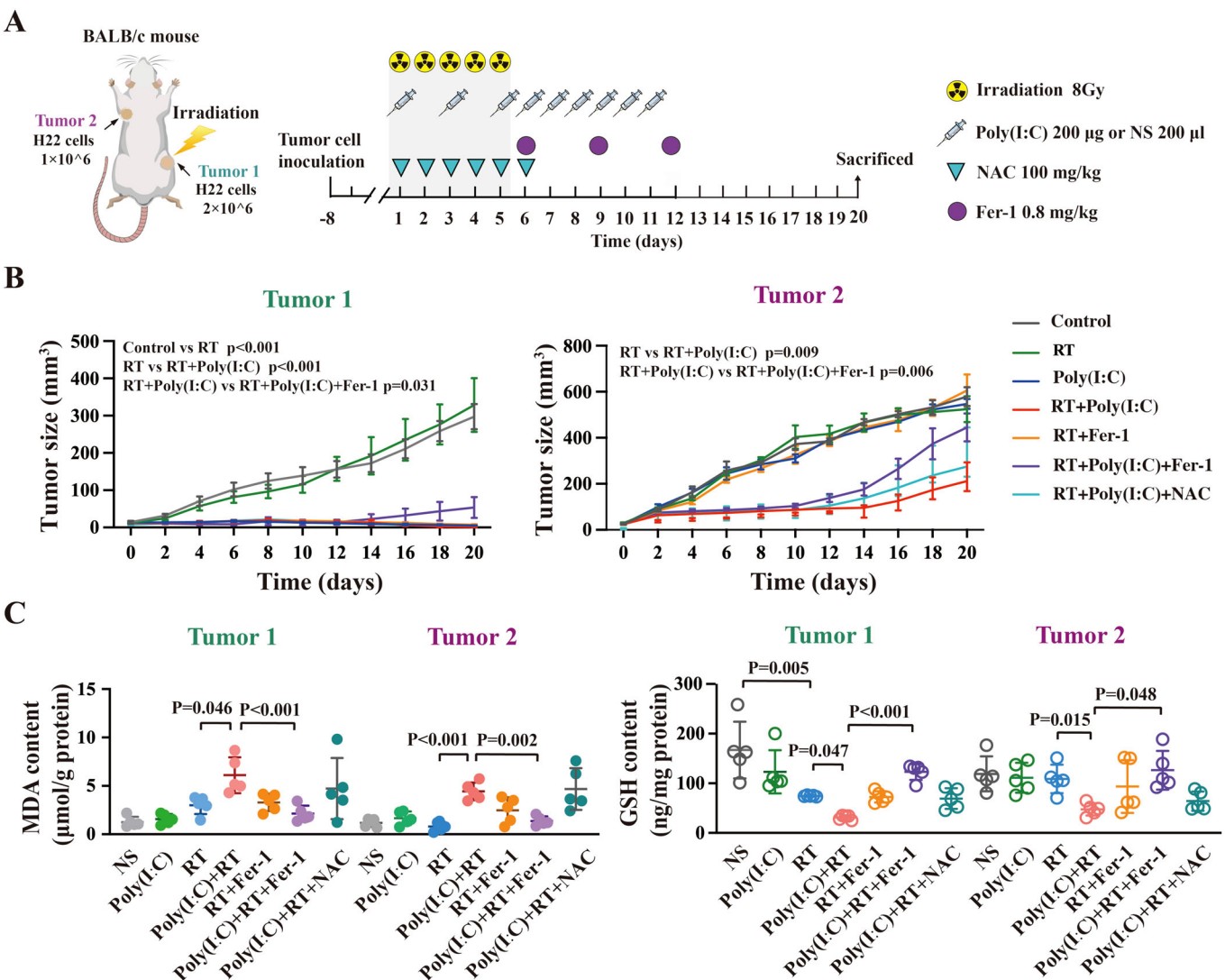

**Figure EV2. Abscopal effect of radiotherapy enhanced by poly(I:C) stimulated tumor ferroptosis.**

(A) Schematic diagram of modeling and therapy of bilateral HCC subcutaneous tumor model in BALB/c mouse. (B) Tumor growth curves of directly irradiated tumor (Tumor1) and distant tumor (Tumor2) in different treated groups ($n = 5$ mice for each group) as indicated. Results are shown as mean ± SEM (error bar), and ANOVA was performed to analyze the differences among groups. (C) Determination of MDA content and GSH content in Tumor1 and Tumor2 collected from different treated groups ($n = 5$ independent samples for each group) as indicated (illustrated by scatterplot). The data represent the means ± SEM (error bar), and ANOVA was performed to analyze the differences among groups.

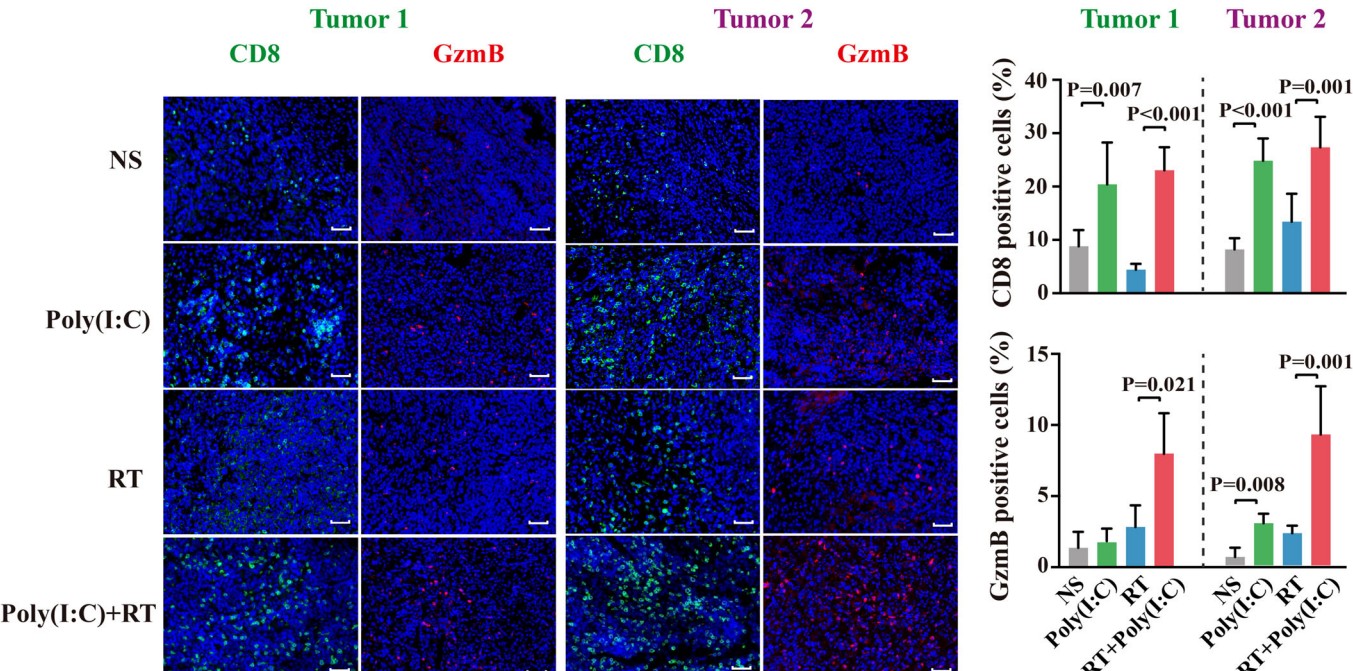

**Figure EV3. Effects of poly(I:C) combined with RT on immune cell infiltration and vascular endothelial chemotaxis in tumors from HCC mice model.**

The representative immunofluorescence image of CD8+ T cells and GzmB in tumor tissues. Scale bars, 100 μm (100×). The proportion of CD8 positive cells and GzmB positive cells are represented by mean ± SEM(error bar) of 5 fields in the tumor tissue samples, and ANOVA was performed to analyze the differences among groups.

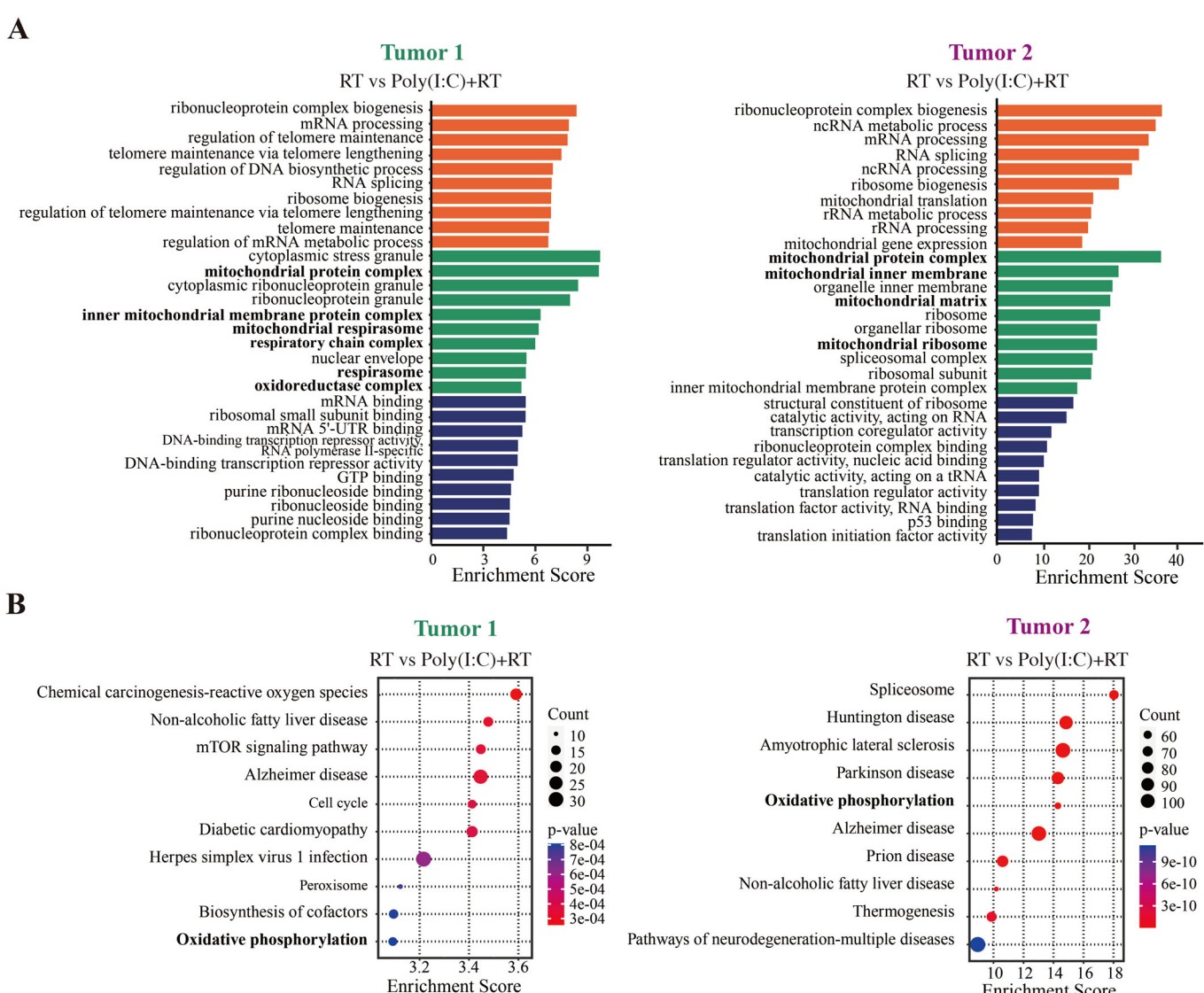

**Figure EV4. Go annotation and pathway analysis of the differentially expressed genes in tumors derived from subcutaneous HCC mouse model receiving different treatments.**

(A) GO analysis histogram. The horizontal axis represents the Enrichment score of the input genes in the Go annotation, while the vertical axis represents the name of Go annotation. Go annotation of differentially expressed genes with top ten Enrichment score covering domains of biological processes (Orange Brown), cellular components (Onion green), and molecular functions (navy blue). (B) Bubble map of the KEGG pathway enrichment analysis. The horizontal axis represents the Enrichment score of the input genes in the pathway, while the vertical axis represents the pathway name.

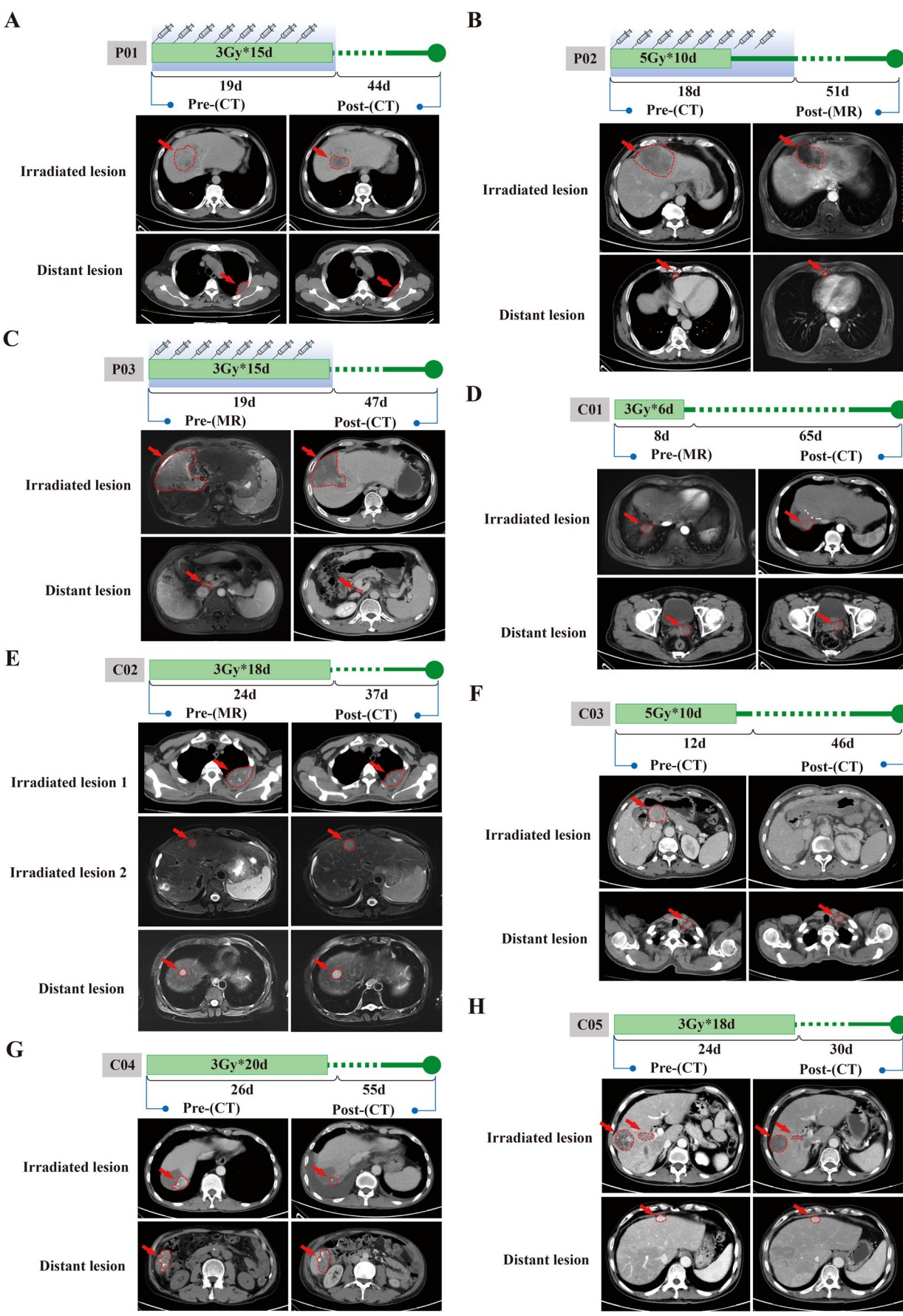

◀ **Figure EV5. Representative MR/CT scans of enrolled patients and control cases.**

Tumor lesions were outlined by red dash circle and marked with red arrows on representative MR/CT scans of enrolled patients P01, P02, P03, and control cases C01, C02, C03, C04, C05. (**A**) Representative imaging scans of patient P01: the irradiated intrahepatic tumor lesion was reduced by 35.1%, while the non-irradiated lymph node metastasis was reduced by 49.0%. (**B**) Representative imaging scans of patient P02: the irradiated tumor located on the left lobe of liver was reduced by 25.7%, while the non-irradiated cardiac phrenic horn lymph node metastasis was reduced by 49.9%. (**C**) Representative imaging scans of patient P03: the total volume of irradiated tumor lesions (located on the right lobe of liver and portal vein) was reduced by 31.2%, while the non-irradiated retroperitoneal lymph node metastasis was reduced by 63.5%. (**D**) Representative imaging scans of patient C01: the irradiated tumor located on right lobe of liver was reduced by 7.3%, while the non-irradiated pelvic lymph node metastasis was reduced by 9.6%. (**E**) Representative imaging scans of patient C02: the irradiated lymph node metastasis located on the 2nd paracostal was reduced by 26.7%, while the non-irradiated intrahepatic tumor was increased by 33.2%. (**F**) Representative imaging scans of patient C03: the irradiated portal lymph node metastasis was eliminated, while the non-irradiated intraperitoneal lymph node metastasis was increased by 2.9%. (**G**) Representative imaging scans of patient C04: the irradiated tumor located on right lobe of liver (near to diaphragmatic surface) was reduced by 54.7%, while the non-irradiated intrahepatic tumor was increased by 8.5%. (**H**) Representative imaging scans of patient C05: the irradiated tumor located on right lobe of liver was reduced by 31.8%, while the non-irradiated tumor located on left lobe of liver was increased by 2.7%.

