## [Peer Review File · EMBO Molecular Medicine]

TLR3 Activation Enhances Abscopal Effect of Radiotherapy in HCC by Promoting Tumor Ferroptosis

Liman Qiu, Hongbing Ji, Kai Wang, Wenhan Liu, Qizhen Huang, Xinting Pan, Honghao Ye, Zhenli Li, Geng Chen, Xiaohua Xing, Xiuqing Dong, Ruijing Tang, Haipo Xu, Jingfeng Liu, Zhixiong Cai, and Xiaolong Liu

Corresponding authors: Xiaolong Liu (liuxl@fjirsm.ac.cn) , Zhixiong Cai (zhixiongcai@fjmu.edu.cn)

Review Timeline:

Submission Date:	28th Oct 23
Editorial Decision:	28th Nov 23
Appeal:	6th Feb 24
Editorial Decision:	19th Mar 24
Revision Received:	3rd Apr 24
Accepted:	9th Apr 24

Editor: Lise Roth

Transaction Report:

28th Nov 2023

Decision on your manuscript EMM-2023-18908

Dear Prof. Liu,

Thank you for the submission of your manuscript to EMBO Molecular Medicine. We have now received the reports from the three reviewers who accepted to evaluate your manuscript.

As you will see from the enclosed reports, the referees acknowledge the potential interest of the findings, however, they also raise a large number of concerns (including, but not limited to, the unclear translational benefits over already established immunotherapies, the limited novelty, missing controls, limited sample size, and discrepancies in the results) and are overall not convinced that the conclusions are supported by the data.

Given the nature of the referees' concerns and the amount of time and work that would be required to address them and considering also that at EMBO Press we encourage one round of revisions only in a reasonable time frame, I am afraid I see little choice but to return the manuscript to you at this point with the decision that we cannot offer to publish it.

I am very sorry to disappoint you in this occasion, and hope that the referees' comments are helpful in your continued work in this area.

With kind regards,

Lise Roth

Lise Roth
Senior Editor
EMBO Molecular Medicine

***** Reviewer's comments *****

Referee #1 (Comments on Novelty/Model System for Author):

Regarding the "medical impacts," it is not clear what additional benefits, considering side effects, poly(I:C) has compared to already established immunotherapy for HCC.

Regarding "novelty," similar experiments were conducted in 2017 by de la Torre et al. (10.2147/JHC.S136652), which the author did not cite or discuss. Why is the current concept presented here considered better than the one applied in 2017?

Referee #1 (Remarks for Author):

The manuscript by Qiu et al. with the title TLR3 Activation Enhances Abscopal Effect of Radiotherapy in HCC by Promoting Tumor Ferroptosis is a very well-executed piece of work that represents a potential combination therapy of radiotherapy and TLR3 activation through poly(I:C). After a thorough examination of this work, I have not found several points of criticism.

Nevertheless, before publication in EMBO, a revised version should be created. My comments are as follows:

The abscopal effects in the introduction are described briefly and incompletely. While it is true that with RT alone, there is only an occasional abscopal response, a significant occurrence is observed with a combination treatment of RT and immunotherapy. The authors should address this in the introduction.

The authors should perform suitable experiments to rule out the possibility that the administration of pentobarbital during the irradiation of animals causes immunomodulation. Pentobarbital is known to specifically influence the function of neutrophilic granulocytes (e.g., DOI:10.1111/j.1525-1489.2001.00079.pp.x). Why was a less burdensome anesthesia with isoflurane not used?

In the last part of your manuscript, the authors describe the effect of the combined application of poly(I:C) with RT in HCC patients. I miss a table with the side effects of the therapy, as is customary in clinical studies. The authors must describe the side effects that could be recorded.

It is also essential to specify the irradiation scheme applied in the clinical study. According to the statement in Figure 7B, this varies. However, it is now clear that not all fractionation schemes and dose concepts in conjunction with immunotherapy lead to desired anti-tumor or abscopal effects. The authors need to highlight this and clearly state which patients were treated with

which irradiation concept.

The discussion needs to be fundamentally improved. While the results of the authors are very impressive, there is a lack of a clear distinction from existing experiments with clinically accepted immunotherapy in HCC. Is poly(I:C) truly better suited to induce significant tumor control or even cure in combination with RT than what has been seen with immune checkpoint inhibitors? What are the possible advantages, if any, of poly(I:C) compared to already established immunotherapy for HCC (e.g. DOI: 10.3748/wjg.v29.i6.1054)?

Referee #2 (Comments on Novelty/Model System for Author):

The models used are sufficient

Referee #2 (Remarks for Author):

In this manuscript, Qiu et al. demonstrated that TLR3 agonist could strengthen the abscopal effect of RT by activating tumour cell ferroptosis in both the murine model and clinical study in n=5 patients with advanced HCC. The study is interesting and comprehensive. As TLR3 agonist is commonly used as an immune adjuvant, combining it with RT shows promising improved outcome for HCC patients. Few comments to improve the manuscript as follows:

1. Fig.2B & 2C, comparison should be done between polyI:C + RT, single treatment and control. Fig. 2F & 2E, comparison should be done between polyI:C + RT + Fer 1 with control.
2. It is not sure why the authors failed to observe the direct apoptosis effect of PolyI:C, which has been shown in previous studies eg. <https://www.tandfonline.com/doi/full/10.1080/15384047.2017.1373220> or <https://www.tandfonline.com/doi/full/10.1080/2162402X.2018.1426434> or <https://pubmed.ncbi.nlm.nih.gov/23197495/>
3. CD103 is not exactly the marker for cDC1 rather markers like Clec9 or XCR1 should be used.
4. Fig. 5B, it is apparent that GzmB+ cells are also increase upon Poly I:C treatment alone but the graph on the right does not quite reflect this.
5. If the authors suggest that the effect of poly(I:C) on irradiation-induced tumor ferroptosis is dependent on the activation of TLR3 signaling of tissue-infiltrating CD8+T cells, then these CD8 T cells must express the TLR3, there is however no data to support expression of TLR3 on CD8+ T cells.
6. Would the authors anticipate the use of polyI:C + RT + checkpoint inhibitor? Could that further enhance the anti-tumour immunity?

Minor comments:

1. The author should also cite <https://pubmed.ncbi.nlm.nih.gov/23197495/> and discuss the potential effect of TLR3 on NK cells. Also it does seem
2. Substantial language editing might be needed to improve the quality of this manuscript e.g. the use of inappropriate descriptions like "luckily".
3. It should be mentioned that polyI:C binds to both TLR3 and MDA5 and from Fig. 3 it seems that the effect is specifically via TLR3 and not MDA5.
4. It is not clear how the authors measure apoptosis or necrosis cell death in Fig. S1
5. It is not clear how the data on Treg, M1 and M2 were derived in Fig. S4A.
6. The data from Fig. S6 should be mentioned in the result section instead.

Referee #3 (Remarks for Author):

The paper presented by Qiu et al., focuses on the involvement of TLR3 in mediating Abscopal Effect of Radiotherapy in HCC by Promoting Tumor Ferroptosis.

Overall, this is an interesting study. Experiment are well done. However, the major limitation of this work is the sample size and the way in which the authors present tumor growth and survival data. Authors should explain how they considered which animals to include in plotting tumor growth and survival curves!

The authors indicate a tumor size of approximately 20mm³, but the authors do not specify whether it is tumor 1 or tumor 2! if it is tumor 1 which is taken into account, then what is the size of tumor 2 on the day of randomization?

Figure 1: the average tumor size (Figure 1D) as well as the individual curves (Fig 1C) are followed until day 18, while on the survival curve, we can clearly see that the animals in the different groups die before the day 18, there is clearly a problem in this figure, it's curious how the authors went about drawing the curves! in any case it is not possible to show an average tumor size up to day 18 and at the same time present a survival curve where the animals die from day 2!!!

Figure2, Authors should show the overall survival.

Figure 3, the authors inject 1×10^6 cells for tumor 1 while in material and method (page 8) and in figure 1 the authors indicate that they inject 1.5×10^6 cells to have tumor1!

All experiments were performed once with a small sample size ($n=3-5$). Experiments should be confirmed.

As a service to authors, EMBO provides authors with the possibility to transfer a manuscript that one journal cannot offer to publish to another EMBO publication. The full manuscript and if applicable, reviewers reports are automatically sent to the receiving journal to allow for fast handling and a prompt decision on your manuscript. For more details of this service, and to transfer your manuscript to another EMBO title please click on Link Not Available

Referee #1:**Comments (Comments on Novelty/Model System for Author):**

Regarding the “medical impacts,” it is not clear what additional benefits, considering side effects, poly(I:C) has compared to already established immunotherapy for HCC. Regarding "novelty," similar experiments were conducted in 2017 by de la Torre et al. (10.2147/JHC.S136652), which the author did not cite or discuss. Why is the current concept presented here considered better than the one applied in 2017?

Response:

Thanks very much. We fully agree with the reviewer’s extremely constructive comments. Regarding the “medical impacts”, poly(I:C) injection has been widely used as the adjuvant treatment for viral keratitis, herpes simplex infection and chronic viral hepatitis in China, with high safety and very low side effects. Meanwhile, judging from clinical trials, when we used it as an adjuvant for neoantigen vaccines or radiotherapy, no patients suffered any obvious toxic side effects. In addition, poly(I:C) combined with RT can exert similar abscopal effects as other immunotherapies such as immune checkpoints, and can be further combined with immune checkpoints to enhance abscopal effects. Importantly, it is very cheap, with the price only one-hundredth of that of immune checkpoint inhibitors, making it very suitable for large-scale clinical promotion, especially in developing countries. Moreover, to better present the clinical translational potential of our study, we summarized the side effects of the patients in the clinical trial in Supplementary Table S5, where only one patient experienced mild treatment-related adverse event (local pain, Grade 1). Thereby, combined with the data in Supplementary Table S6, the clinical application of radiotherapy combined with poly(I:C) is safe and well tolerated. The relevant descriptions are added in the “Results” section as follows: “**Significantly, no obvious treatment related adverse events were reported and routine blood/biochemical tests also showed no obvious abnormalities in those 5 enrolled patients during the whole trial period (Supplementary Table S5 and Supplementary Table S6)**”, and the “Discussion” section as follows: “**In terms of therapeutic efficacy, through animal models and prospective clinical studies, we have found the same phenomenon, that is,**

when the HCC mice models or HCC patients are further subcutaneously injected with poly(I:C) during RT for the target lesion, the abscopal effect can be significantly enhanced. Regarding safety, poly(I:C) has been widely in clinical with high safety and very low side effects. Moreover, our studies also demonstrated that poly(I:C) was a safe and effective adjuvant for tumor neoantigen vaccine or combining with RT (Cai *et al.*, 2021). Importantly, it is very cheap, with the price only one-hundredth of that of immune checkpoint inhibitors, making it very suitable for large-scale clinical promotion, especially in developing countries. Therefore, although the enrollment of patients is limited in this study, RT combined with poly(I:C) has proven to be a safety and feasible strategy in HCC. However, we also found that RT combined with poly(I:C) also cannot maintain this phenomenon for a long time, and new lesions will still occur later. This may be mainly because the highly expressed immune checkpoint receptors such as PD-L1 in the immunosuppressive microenvironment of HCC could bind to the PD-1 receptor of T cells infiltrating into tumor tissue, thus leading to T cell dysfunction or depletion and the subsequent occurrence of new tumor lesions. Our preliminary experiments in HCC mouse models also showed that further combination of α -PD1 on the basis of poly(I:C) and RT treatment can further enhance and prolong the efficacy and duration of abscopal effect (Supplementary FigureS2G), which can be served as a further potential improvement scheme, but the effect remains to be further explored, and the related toxicity still need to be further thoroughly investigated.”

Regarding "novelty", although our study and the study from de la Torre et al both used a combination strategy of RT combined with TLR3 agonist [poly-ICLC or poly(I:C)], there are big differences between the two in terms of specific implementation. In the study by de la Torre et al in 2017, they used intra-tumoral injection of the TLR3 agonist poly-ICLC via ultrasound-guiding after low local nonlethal radiation (2.5 Gy/fraction, a total dose of 22.5 Gy) and local regional treatment (TAE or TACE) to treat advanced HCC patients. Here, we largely optimized the combined treatment plan of RT and poly(I:C). Regarding the irradiation dose for HCC patients, our study employs a therapeutic dosage (3~5 Gy/fraction, a

total dose of 40~50 Gy) which is more easier by the tumor cells to generate tumor associated antigens or mutational neoantigens (4 Gy or 9 Gy for one time, doi: 10.1073/pnas.2102611118) instead of relatively low dose employed by de la Torre et al. (2.5 Gy/fraction, a total dose of 22.5 Gy). Secondly, for the method of poly(I:C) injecting, we adopted a non-invasive approach by performing subcutaneous injection of poly(I:C) during the entire RT cycle of the patient, instead of invasive intra-tumoral injection into loin regio by de la Torre et al, thus improving the clinical feasibility and reducing the risk of complication to increase the acceptance of patients. Thirdly, for the time of poly(I:C) injecting, instead of twice a week injection after RT and local regional treatment for the next 4 weeks, we performed poly(I:C) injection in every two days during entire RT cycle, which could help to make full use of tumor associated antigens released from RT treatment in real time to promote antigen presentation and CD8⁺ T cell activation. Finally, the molecular mechanism of RT combined with poly(I:C) is largely unclear in the study by de la Torre et al, while we performed a comprehensive study of its potential anti-tumor mechanisms and reported for the first time that poly(I:C) could directly act on CD8⁺ T cells to exert abscopal effects via tumor ferroptosis.

In addition, we are deeply grateful for the reviewer's extremely valuable suggestion. According to the reviewer's comment, we cited the relevant paper published in 2017 by de la Torre et al, which actually helped us a lot. The relevant introduction and the further discussion were added in "Introduction" section as follows: "A phase I clinical trial by the combination of intra-tumoral injection of the TLR3 agonist poly-ICLC after local radiation at relatively low dose (2.5 Gy/fraction, a total dose of 22.5 Gy) and local regional therapy (TAE or TACE) to treat advanced HCC patients, has proved to be safe and tolerable (de la Torre *et al.*, 2017). However, the abscopal effect has been only observed in a small number of patients and the underlying molecular mechanism remains unclear (de la Torre *et al.*, 2017)." and "Discussion" section as follows: "Although the treatment regimen of RT combined with poly(I:C) has been proven to be feasible in patients with advanced HCC, the abscopal effect is not significant and the mechanism is unclear (de la Torre *et al.*, 2017). Here, we

optimized the combined treatment plan of RT and poly(I:C). For RT in HCC patients, we use therapeutic dose (3~5 Gy/fraction, a total dose of 40~50 Gy) to treat the targeted tumor lesion, which could release more tumor associated antigens for immune system activation; secondly, in terms of the method and time of poly(I:C) injecting, subcutaneous injection of poly(I:C) was performed in every two days in the entire RT cycle, which could help to make full use of tumor associated antigens released from RT treatment in real time to promote antigen presentation and CD8⁺ T cell activation. Meanwhile, because this treatment plan does not use invasive treatment measures, it is clinically operable and has a high degree of patient acceptance.”

Comments (Remarks for Author):

The manuscript by Qiu et al. with the title TLR3 Activation Enhances Abscopal Effect of Radiotherapy in HCC by Promoting Tumor Ferroptosis is a very well-executed piece of work that represents a potential combination therapy of radiotherapy and TLR3 activation through poly(I:C). After a thorough examination of this work, I have not found several points of criticism. Nevertheless, before publication in EMBO, a revised version should be created.

Response:

Thanks very much. We sincerely appreciate the reviewer’s recognition of the potential clinical value of this study, and your positive comments on the quality of the data presented. These comments support the significance of our research. The arisen issues have been carefully addressed point by point as follows.

Question 1: The abscopal effects in the introduction are described briefly and incompletely. While it is true that with RT alone, there is only an occasional abscopal response, a significant occurrence is observed with a combination treatment of RT and immunotherapy. The authors should address this in the introduction.

Response: We sincerely appreciate the extremely helpful suggestion. Following reviewer’s suggestion, we provided related description in “Introduction” section as follows: “RT combined with immune checkpoint inhibitors has been shown to increase the rate of abscopal effect in patients with advanced tumors (Ozpiskin *et al*,

2019; Zhao & Shao, 2020) . However, the rate of enhanced abscopal effects is still unsatisfactory, and a large number of patients are still unable to accept or tolerate it due to the relatively large side effects and expensive drugs.”

Question 2: The authors should perform suitable experiments to rule out the possibility that the administration of pentobarbital during the irradiation of animals causes immunomodulation. Pentobarbital is known to specifically influence the function of neutrophilic granulocytes (e.g., DOI:10.1111/j.1525-1489.2001.00079.pp.x). Why was a less burdensome anesthesia with isoflurane not used?

Response: We are deeply grateful for your extremely valuable suggestion. As you mentioned, isoflurane was a less burdensome anesthesia. We have actually tried to irradiate mice after transient inhalation anesthesia with isoflurane. However, we found that this did not ensure that the mice at the appropriate depth of anesthesia to be positioned and irradiated by the experimenter. Some of the mice struggled during the experiment, which causing pain to the mice and unable to ensure that the tumors of the mice could be exposed to the irradiated area during the irradiation process. We have also considered allowing mice to inhale isoflurane continuously at appropriate concentrations to maintain a safe and sufficient depth of anesthesia for mice (mice needed to be placed in a device that provided a relatively confined space). However, due to the potential interference of the radiation caused by the additional device, we ultimately did not use such anesthesia. Thus, in order to alleviate the pain of the mice and ensure the progress of the experiment, we finally chose 50 mg/kg pentobarbital sodium to anesthetize the mice, so as to ensure that the mice receive treatment in a quiet state and wake up after 20 minutes of treatment. Furthermore, although pentobarbital is known to specifically influence the function of neutrophilic granulocyte, the sequencing data (analyzed by ImmuCellAI) from our mouse tumor tissues suggested that the neutrophils infiltration scores of unanesthetized mice [mice in NS group and poly(I:C) group] were not significantly different from those of mice anesthetized with pentobarbital sodium [mice RT group and poly(I:C)+RT group], as shown in Figure1. Therefore, the effect of pentobarbital sodium on neutrophil is

relatively limited in our current study.

Figure 1. Infiltration score of neutrophil in tumor tissues according to RNA sequencing data analyzed by ImmuCellAI.

Question 3: In the last part of your manuscript, the authors describe the effect of the combined application of poly(I:C) with RT in HCC patients. I miss a table with the side effects of the therapy, as is customary in clinical studies. The authors must describe the side effects that could be recorded.

Response: We are grateful for the reviewer’s extremely valuable guidance. To address it, the treatment-related adverse events were showed in Supplementary Table S5, and the related description was supplemented in “Results” section of “Poly(I:C) improves radiotherapy efficacy in patients with advanced HCC” as follows: “Significantly, no obvious treatment related adverse events were reported and routine blood/biochemical tests also showed no obvious abnormalities in those 5 enrolled patients during the whole trail period (Supplementary Table S5, Supplementary Table S6).”

Question 4: It is also essential to specify the irradiation scheme applied in the clinical study. According to the statement in Figure 7B, this varies. However, it is now clear that not all fractionation schemes and dose concepts in conjunction with immunotherapy lead to desired anti-tumor or abscopal effects. The authors need to highlight this and clearly state which patients were treated with which irradiation concept.

Response: Thanks very much for the reviewer’s extremely constructive comments. Following reviewer’s suggestion, we provided the detail of irradiation concept in

“Method” section of “Clinical trial design and intervention” as follows: “Patient's radiotherapy scheme (3~5 Gy/fraction, a total dose of 40~50 Gy) is a tailor-made customization based on the patient's tumor size and location by the radiologist.”

Question 5: The discussion needs to be fundamentally improved. While the results of the authors are very impressive, there is a lack of a clear distinction from existing experiments with clinically accepted immunotherapy in HCC. Is poly(I:C) truly better suited to induce significant tumor control or even cure in combination with RT than what has been seen with immune checkpoint inhibitors? What are the possible advantages, if any, of poly(I:C) compared to already established immunotherapy for HCC (e.g. DOI: 10.3748/wjg.v29.i6.1054)?

Response: We sincerely appreciate the extremely valuable suggestion. According to the reviewer's suggestion, we have revised the discussion to highlight the advantages of poly(I:C) in combination with RT in the treatment of HCC by comparing the differences in efficacy and tolerability between poly(I:C) and established immunotherapy for HCC, as follows: “Presently, RT combined with immune checkpoint inhibitors (such as PD-1 antibody or CTLA-4 antibody) has been shown to induce about 29% rate of abscopal effect in patients with advanced tumors; nevertheless, the incidence of abscopal effect is still unsatisfactory, which may be caused by lack of tumor infiltrating activated T cells and further make it impossible to initiate an adequate immune response (Ozpiskin *et al.*, 2019; Zhao & Shao, 2020). In addition, the side effects of immunotherapy was nonnegligible (Mandlik *et al.*, 2023), which limits the combined application of radiotherapy and immunotherapy. Although the treatment regimen of RT combined with poly(I:C) has been proven to be feasible in patients with advanced HCC, the abscopal effect is not significant and the mechanism is unclear (de la Torre *et al.*, 2017). Here, we optimized the combined treatment plan of RT and poly(I:C). For RT in HCC patients, we use therapeutic dose (3~5 Gy/fraction, a total dose of 40~50 Gy) to treat the targeted tumor lesion, which could release more tumor associated antigens for immune system activation; secondly, in terms of the method and time of poly(I:C) injecting, subcutaneous injection of poly(I:C) was performed in every two days in the entire RT cycle, which could help to

make full use of tumor associated antigens released from RT treatment in real time to promote antigen presentation and CD8⁺ T cell activation. Meanwhile, because this treatment plan does not use invasive treatment measures, it is clinically operable and has a high degree of patient acceptance. In terms of therapeutic efficacy, through animal models and prospective clinical studies, we have found the same phenomenon, that is, when the HCC mice models or HCC patients are further subcutaneously injected with poly(I:C) during RT for the target lesion, the abscopal effect can be significantly enhanced.

Regarding safety, poly(I:C) has been widely in clinical with high safety and very low side effects. Moreover, our studies also demonstrated that poly(I:C) was a safe and effective adjuvant for tumor neoantigen vaccine or combining with RT (Cai *et al.*, 2021). Importantly, it is very cheap, with the price only one-hundredth of that of immune checkpoint inhibitors, making it very suitable for large-scale clinical promotion, especially in developing countries. Therefore, although the enrollment of patients is limited in this study, RT combined with poly(I:C) has proven to be a safety and feasible strategy in HCC. However, we also found that RT combined with poly(I:C) also cannot maintain this phenomenon for a long time, and new lesions will still occur later. This may be mainly because the highly expressed immune checkpoint receptors such as PD-L1 in the immunosuppressive microenvironment of HCC could bind to the PD-1 receptor of T cells infiltrating into tumor tissue, thus leading to T cell dysfunction or depletion and the subsequent occurrence of new tumor lesions. Our preliminary experiments in HCC mouse models also showed that further combination of α -PD1 on the basis of poly(I:C) and RT treatment can further enhance and prolong the efficacy and duration of abscopal effect (Supplementary FigureS2G), which can be served as a further potential improvement scheme, but the effect remains to be further explored, and the related toxicity still need to be further thoroughly investigated.”

Referee #2

Comments (Comments on Novelty/Model System for Author):

The models used are sufficient.

Response: Thank you very much for the reviewers' recognition of the application of our research model.

Comments (Remarks for Author):

In this manuscript, Qiu et al. demonstrated that TLR3 agonist could strengthen the abscopal effect of RT by activating tumor cell ferroptosis in both the murine model and clinical study in n=5 patients with advanced HCC. The study is interesting and comprehensive. As TLR3 agonist is commonly used as an immune adjuvant, combining it with RT shows promising improved outcome for HCC patients. Few comments to improve the manuscript as follows:

Response: Thanks very much. We sincerely appreciate the reviewer's recognition. TLR3 agonist in combination with radiotherapy is indeed a safe and effective potential treatment strategy for patients with advanced HCC.

Question 1: Fig.2B & 2C, comparison should be done between polyI:C + RT, single treatment and control. Fig. 2F & 2E, comparison should be done between polyI:C + RT + Fer 1 with control.

Response: We sincerely appreciate the extremely helpful suggestion. Following the reviewer's suggestion, the corresponding P-values have been annotated in Figure2. As shown in Figure 2B and 2C, poly(I:C) plus RT treatment significantly promoted the MDA accumulation and GHS consumption compared with treatment of NS or poly(I:C) alone in both Tumor1 and Tumor2 [MDA accumulation of Tumor1: NS vs Poly(I:C)+RT, $P < 0.001$; Poly(I:C) vs Poly(I:C)+RT, $P < 0.001$. MDA accumulation of Tumor2: NS vs Poly(I:C)+RT, $P = 0.004$; Poly(I:C) vs Poly(I:C)+RT, $P = 0.021$. GHS consumption of Tumor1: NS vs Poly(I:C)+RT, $P < 0.001$; Poly(I:C) vs Poly(I:C)+RT, $P < 0.001$. GHS consumption of Tumor2: NS vs Poly(I:C)+RT, $P < 0.001$; Poly(I:C) vs Poly(I:C)+RT, $P < 0.001$]. As shown in Figure 2E, tumor growth of Tumor1 in poly(I:C)+RT+Fer-1 treated mice was slower than that of NS treated mice [NS vs Poly(I:C)+RT+Fer-1, $P < 0.001$], while there was no significant difference in tumor growth of Tumor2 between poly(I:C)+RT+Fer-1 treated mice and NS treated mice.

Figure 2 (**Figure 2 in manuscript**). Determination of MDA content (B) and GSH content (C) in Tumor1 and Tumor2 in mice with different treatment as indicated (illustrated by scatterplot). (E) Tumor growth curves of Tumor1 and Tumor2 in different treated group as indicated. Results are shown as mean \pm SEM. (F) Determination of MDA content (left panel) and GSH content (right panel) in Tumor1 and Tumor2 in different treated group as indicated (illustrated by scatterplot).

Question 2: It is not sure why the authors failed to observe the direct apoptosis effect of PolyI:C, which has been shown in previous studies eg. <https://www.tandfonline.com/doi/full/10.1080/15384047.2017.1373220> or <https://www.tandfonline.com/doi/full/10.1080/2162402X.2018.1426434> or <https://pubmed.ncbi.nlm.nih.gov/23197495/>

Response: We sincerely appreciate the reviewer's enlightening comment. In fact, our sequencing data of mouse tumor tissue demonstrated that poly(I:C) significantly increased apoptosis enrichment score of Tumor1 (as showed in Figure2A, $P < 0.05$). However, no significant regulatory effect of poly(I:C) on tumor cell apoptosis was observed by Annexin-V assay, which is inconsistent with previous reports that poly(I:C) promotes tumor cell apoptosis. We understand that there are two factors: On

the one hand, poly(IC) administration dose and site used in our study are different from previous studies, which may not have an equivalent apoptosis regulation effect; On the other hand, radiotherapy can induce strong apoptosis of tumor cells, resulting in the weaker promoting effect of poly(I:C) on apoptosis was not detected when combined with radiotherapy.

Question 3: CD103 is not exactly the marker for cDC1 rather markers like Clec9 or XCR1 should be used.

Response: We sincerely appreciate the extremely valuable suggestion. Following the reviewer's suggestion, We identified cDC1 with CD45⁺CD11c⁺CLEC9⁺ cell (instead of CD11c⁺CD103⁺ cell) and investigated the effect of poly(I:C) combined with radiotherapy on intratumoral cDC1 infiltration by tissue multicolor immunofluorescence (Figure 3). The results were supplemented in the "Results" section of "Poly(I:C) promotes DC uptake of neoantigens released from irradiated-tumor tissues" as follows: "We further analyzed the number of cDC1s (marked with CD45⁺CD11c⁺CLEC9⁺) in Tumor1 by multicolor immunofluorescence, and the results showed the cell density of cDC1s in mice treated with poly(I:C) plus RT was 5.86-fold higher than that in mice treated with RT alone [RT 645.03±71.47 vs Poly(I:C)+RT 1833.18±177.46; P<0.001, Figure 4C]. Moreover, poly(I:C) combined with RT significantly increased the IL-12 (cytokine associated with enhanced DC priming) expression of intratumoral cDC1s, compared with RT alone [IL12⁺ cDC1s: RT 352.63±27.92 vs Poly(I:C)+RT 743.51±74.32; P<0.001] (Figure 4C)."

Figure 3 (Figure 4C in manuscript). The representative immunofluorescence image of

CD45⁺CD11c⁺CLEC9⁺ cDC1s in Tumor1 of different treated group (left panel), and the cell density of cDC1s (CD45⁺CD11c⁺CLEC9⁺ cells) and IL12⁺ cDC1s were shown in statistical scatter plots (n=3 per group, right panel). Scale bars, 20 μ m (400 \times). The cell density are represented by mean \pm SEM of entire section field of the tumor tissue samples.

Question 4: Fig. 5B, it is apparent that GzmB⁺ cells are also increase upon Poly I:C treatment alone but the graph on the right does not quite reflect this.

Response: We sincerely appreciate the reviewer's helpful comment. GzmB⁺ cells are significantly increased upon poly(I:C) treatment compared with NS treatment in Tumor2 (P=0.008) but not in Tumor1 (P>0.05) (Figure 4). However, we missed the p-value that should have been marked on the graph. The revised graph was moved to Supplementary FigureS4C.

Figure 4 (Supplementary FigureS4C in manuscript). The representative immunofluorescence image of CD8⁺ T cells and GzmB in tumor tissues. Scale bars, 100 μ m (100 \times). The proportion of CD8 positive cells and GzmB positive cells are represented by mean \pm SEM of 5 fields in the tumor tissue samples.

Question 5: If the authors suggest that the effect of poly(I:C) on irradiation-induced tumor ferroptosis is dependent on the activation of TLR3 signaling of tissue-infiltrating CD8⁺T cells, then these CD8 T cells must express the TLR3, there is however no data to support expression of TLR3 on CD8⁺ T cells.

Response: Thanks very much for the reviewer's extremely constructive comments. TLR3 is indeed expressed on CD8⁺ T cells, which is confirmed by

immunofluorescence co-localization of TLR3 and CD8 in our study (Figure 5A). Significantly, we also found that poly(I:C) can promote TLR3⁺CD8⁺ T cell infiltration to both tumor1 and tumor2 when combined with RT (Figure 5B). The result was supplemented in the “Results” section of “The promoting effect of poly(I:C) on irradiation-induced tumor ferroptosis is dependent on the activation of TLR3 signaling of tissue-infiltrating CD8⁺ T cells” as follows: “Likewise, we also found that poly(I:C) can promote TLR3⁺CD8⁺ T cell infiltration to both Tumor1 and Tumor2 when combined with RT (Figure 5B, Tumor1: RT vs Poly(I:C)+RT, P<0.001; Tumor2: RT vs Poly(I:C)+RT, P<0.001). ”

Figure 5. (A). The representative immunofluorescence colocalization image of TLR3⁺CD8⁺ T cells in tumor tissues. The white arrows indicate TLR3⁺CD8⁺ T cells. Scale bars, 10 μ m (1000 \times). (B, **Figure 5B in manuscript**). The representative immunofluorescence image of TLR3⁺CD8⁺ T cells and GzmB⁺CD8⁺ T cells in tumor tissues(left panel), and the cell density of TLR3⁺CD8⁺ T cells and GzmB⁺CD8⁺ T cells were shown in statistical scatter plots (right panel). Scale bars, 20 μ m (400 \times). The cell density are represented by mean \pm SEM of entire section field of the tumor tissue samples.

Question 6: Would the authors anticipate the use of polyI:C + RT + checkpoint inhibitor? Could that further enhance the anti-tumour immunity?

Response: We sincerely appreciate the extremely valuable suggestion. Following the review's suggestion, we performed the preliminary experiments in HCC mouse models to evaluate the efficacy of poly(I:C) plus RT and anti-PD-1 treatment. The results showed that further combination of α -PD1 on the basis of poly(I:C) and RT treatment can further enhance and prolong the efficacy and duration of abscopal effect (Figure 6), which can be served as a further potential improvement scheme. To address this, we have provided a detailed description in "Discussion" section as follows: "Our preliminary experiments in HCC mouse models also showed that further combination of α -PD1 on the basis of poly(I:C) and RT treatment can further enhance and prolong the efficacy and duration of abscopal effect (Supplementary FigureS2G), which can be served as a further potential improvement scheme, but the effect remains to be further explored, and the related toxicity still need to be further thoroughly investigated."

Figure 6. (Supplementary FigureS2G in manuscript) Tumor growth curves of directly irradiated tumor (Tumor1) and distant tumor (Tumor2) in different treated groups as indicated. Results are shown as mean \pm SEM.

Minor comments:

Question 1: The author should also cite <https://pubmed.ncbi.nlm.nih.gov/23197495/> and discuss the potential effect of TLR3 on NK cells. Also it does seem

Response: Thanks very much. In the revised manuscript, we added discussion of the potential effect of TLR3 on NK cells and cite <https://pubmed.ncbi.nlm.nih.gov/23197495/> in "Discussion" section as follows: "It is worth mentioning that previous studies have confirmed TLR3 activation

promoted NK cell activation and cytotoxicity in vitro (Chew *et al*, 2012), thus, it will also be interesting to investigate whether poly(I:C) also exerts anti-tumor effects by activating the TLR3 receptor of NK cells.”

Question 2: Substantial language editing might be needed to improve the quality of this manuscript e.g. the use of inappropriate descriptions like “luckily”.

Response: Thanks very much. In the revised manuscript, we have corrected this inappropriate description and polished the language throughout the text.

Question 3: It should be mentioned that polyI:C binds to both TLR3 and MDA5 and from Fig. 3 it seems that the effect is specifically via TLR3 and not MDA5.

Response: Thanks very much for the reviewer’s extremely constructive comments. As mentioned by the reviewer, poly(I:C) could activate pattern recognition receptors (PRRs) in the innate immune system, including MDA5 and TLR3. However, MDA5 and TLR3 are not necessarily equally important in tumor immunoregulation of poly(I:C). For example, Chin *et al.* found that TLR3, rather than MDA5, supports a major contribution in tumor growth inhibited by poly(I:C) (doi: 10.1158/0008-5472.CAN-09-1162.). Moreover, Yoshida *et al.* found that tumor growth suppression induced by the therapy of poly(I:C) combined with radiotherapy, was largely abrogated in TLR3-deficient mice with minimal effect in MAVS (downstream adaptor protein of MDA5)-deficient animal, suggesting that the promotion of radiotherapy effect by poly(I:C) was mainly dependent on TLR3 signal, instead of MDA5 (DOI: 10.1111/cas.13543). In our study, through TLR3 knockout experiments, we also proved that poly(I:C) mainly promoted the abscopal effect of RT depending on TLR3 signal (Fig. 3), although the effects of MDA5 cannot be completely rule out.

Question 4: It is not clear how the authors measure apoptosis or necrosis cell death in Fig. S1.

Response: Thanks very much. we measure represented necrosis cells and late apoptotic cells by Annexin V⁺/PI⁺ cells, which was described in legend of Supplementary FigureS1 as follows: “Annexin V⁻/PI⁻ cells were survival cells, Annexin V⁺/PI⁻ cells were defined as early apoptotic cells, Annexin V⁺/PI⁺ cells

represented necrosis cells and late apoptotic cells.”.

Question 5: It is not clear how the data on Treg, M1 and M2 were derived in Fig. S4A.

Response: Thanks very much. The infiltration score of Treg, M1 and M2 showed in Fig. S4A were derived from RNA sequencing data analyzed by ImmuCellAI. In order to make the presentation clearer, we have added relevant information in the “Supplementary Method” section of “Bulk RNA-seq and data analysis” as follows: “The infiltration scores of multiple immune cells were estimated by the online analysis site ImmuCellAI (<http://bioinfo.life.hust.edu.cn/ImmuCellAI#!/ImmuCellAI>), which is a tool to estimate the abundance of 36 immune cells, including 16 T cells (CD8⁺ T cells, CD4⁺ T cells, Treg, etc.), 2 Macrophages (M1, M2), B cell, NK cell, Monocytes, Neutrophils, etc.”

Question 6: The data from Fig. S6 should be mentioned in the result section instead.

Response: Thanks very much. The data from Fig.S6 have been mentioned in the “Results” section of “Poly(I:C) promotes T cell infiltration in tumor tissues during RT” as follows: “Moreover, GO&Pathway analysis also revealed that poly(I:C) plus RT treatment resulted in significant enrichment of oxidative phosphorylation signaling pathway and cellular mitochondrial structure-related cellular components (mitochondria, as the primary regulator of oxidative phosphorylation, could regulate the sensitization of cells to ferroptosis (Wang et al, 2020),(Bock & Tait, 2020) in both Tumors (especially in Tumor2), when compared with RT alone (Supplementary FigureS5).”

Referee #3

Comments (Remarks for Author):

The paper presented by Qiu et al., focuses on the involvement of TLR3 in mediating Abscopal Effect of Radiotherapy in HCC by Promoting Tumor Ferroptosis. Overall, this is an interesting study. Experiment are well done. However, the major limitation of this work is the sample size and the way in which the authors present tumor growth and survival data. Authors should explain how they considered which animals to

include in plotting tumor growth and survival curves!

Response: Thanks very much. We sincerely appreciate the reviewer's recognition that this is an interesting study. The major limitation of this work raised by the reviewers (the sample size and the way in which the authors present tumor growth and survival data) will be addressed one by one below.

Question 1: The authors indicate a tumor size of approximately 20mm^3 , but the authors do not specify whether it is tumor 1 or tumor 2! if it is tumor 1 which is taken into account, then what is the size of tumor 2 on the day of randomization?

Response: Thanks very much. On the day of randomization, Tumor1 and Tumor2 were $\sim 20\text{mm}^3$ and $\sim 25\text{mm}^3$, respectively. We have added relevant information in the "Method" section of "In vivo tumor irradiation experiments" as follows: "Seven days after the inoculation, the tumors were established (Tumor1 and Tumor2 were $\sim 20\text{mm}^3$ and $\sim 25\text{mm}^3$, respectively.) and the mice started to receive corresponding treatment."

Question 2: Figure 1: the average tumor size (Figure 1D) as well as the individual curves (Fig 1C) are followed until day 18, while on the survival curve, we can clearly see that the animals in the different groups die before the day 18, there is clearly a problem in this figure, it's curious how the authors went about drawing the curves! in any case it is not possible to show an average tumor size up to day 18 and at the same time present a survival curve where the animals die from day 2!!!

Response: Thanks very much. However, we consider that the reviewer may have misinterpreted the data in Figure1E, which is a curve of distant progression survival rather than overall survival. Even if the nonirradiated distant tumor (Tumor2) in the mice progressed on day2 (the tumor reached 300mm^3 , but the mice did not die), we could continue to record the tumor size and plot the growth curves until the mice were sacrificed (day20). Thus, the average tumor size (Figure 1D) and the individual curves (Fig 1C) are followed until day 18.

Question 3: Figure2, Authors should show the overall survival.

Response: Thanks very much. In the experiment shown in Figure 2, we focused more on the progression of unirradiated distal tumor than overall survival. In detail, to

determine whether poly(I:C) could improve the efficiency of radiotherapy by enhancing tumor ferroptosis, we investigate the effect of ferroptosis inhibitor Fer-1 on the effect of poly(I:C) enhanced abscopal effect and the promoted tumor cell ferroptosis. The abscopal effect was directly reflected by the progression of unirradiated distant tumors, rather than overall survival in mice. Therefore, we did not examine the overall survival time, but monitored the growth of distant tumors of mice in Figure 2.

Question 4: Figure 3, the authors inject 1×10^6 cells for tumor 1 while in material and method (page 8) and in figure 1 the authors indicate that they inject 1.5×10^6 cells to have tumor 1!

Response: Thanks very much for reading our manuscript carefully and pointing out this mistake. After checking, we found that the number of injected cells in Tumor 1 was 1×10^6 , which was marked incorrectly in Figure 1A (1.5×10^6). We have corrected this mistake.

Question 5: All experiments were performed once with a small sample size ($n=3-5$). Experiments should be confirmed.

Response: Thanks very much for this extremely helpful suggestion. To address this, the results of tumor growth curves modeled by C57BL/6 mice from different batches of experiments were jointly presented in Figure 1C~E ($n=10$ per group, the experiment was independently repeated twice, each time with 5 mice per group). Actually, we also used different mouse models to demonstrate the same conclusion, therefore, it seems only 3-5 mice per time. In order to provide more valuable biological replication and sufficient sample size for the conclusions of this study, we also used BALB/c mice ($n=5$ per group) to validate the results of key experiments, including tumor growth curves, detection of tumor ferroptosis, and in vivo block of CD8⁺ T cell functions, etc (Supplementary Figure S3B~D). Moreover, TLR3 knock-out mice were used to further confirm the criticality of TLR3 signaling in poly(I:C) promoting tumor ferroptosis and enhancing the abscopal effect of radiotherapy. Although it was not possible to obtain enough TLR3 knockout mice in a short period of time, but we did observe data that supported the experimental conclusions from the other two

mouse models. Overall, our study used sufficient animal models and number of mice, which provides a reliable biological replication to solidify our experimental conclusions.

19th Mar 2024

Dear Prof. Liu,

Thank you for your appeal and for submitting your revised study. As the manuscript was initially rejected based on the estimated amount of work needed to address all referees' concerns (and not based on scope or novelty), and since you provided a fully revised manuscript, we decided to send it back to the original reviewers. Please accept my apologies for the delay in getting back to you as one referee needed more time to complete his/her review. We have now received the three reports, and as you will see below, the referees are mostly satisfied with the revisions. I will therefore be able to accept your manuscript once the following points will be addressed:

1/ Referees: Please address the remaining concerns from referees #1 (discuss the clinical relevance of the PolyIC injection) and #2.

2/ Manuscript text:

- Please remove the red font, and only keep in track changes mode any new modification.
- Please provide institutional email addresses for both corresponding authors, both in the submission system and in the manuscript.
- We can accommodate a maximum of 5 keywords, please adjust accordingly.
- Please re-order and rename the manuscript sections as outlined in the author guidelines: Abstract, Introduction, Results, Discussion, Methods, Acknowledgements, Disclosure and competing interests' statement, References, Figure legends.
- Please remove the synopsis paragraph from the manuscript.
- Materials and Methods:
 - o Please include the supplementary method in the main manuscript text.
 - o Mice: please indicate the housing and husbandry conditions for the mice.
 - o Patients: please include the full sentence that the experiments conformed to the principles set out in the WMA Declaration of Helsinki and the Department of Health and Human Services Belmont Report.
 - o Antibodies: please provide references and dilutions/concentrations.
 - o Statistics: please include a statement on sample size, blinding, randomization.
- Please correct the format of your Data Availability section: As per our guidelines, large-scale datasets, sequences, and computational models should be deposited in one of the relevant public databases prior to submission. Accession codes should be included in a "Data Availability" section at the end of Materials & Methods (suggested wording: "The [protein interaction | microarray | mass spectrometry] data from this publication have been deposited to the [name of the database] database [URL] and assigned the identifier [accession | permalink | hashtag].")
- Please merge the funding with the Acknowledgements section and enter the same information in the submission system.
- Author contributions: CRediT has replaced the traditional author contributions section because it offers a systematic machine-readable author contributions format that allows for more effective research assessment. Please remove the Authors Contributions from the manuscript and use the free text boxes beneath each contributing author's name in our system to add specific details on the author's contribution. More information is available in our guide to authors.
- "Competing interests" should be renamed "Disclosure statement and competing interests": We updated our journal's competing interests policy in January 2022 and request authors to consider both actual and perceived competing interests. Please review the policy <https://www.embopress.org/competing-interests> and update your competing interests if necessary.

3/ Figures and Appendix:

- Please upload your figures as TIFF, EPS or PDF format
 - We replaced Supplementary Information with Expanded View (EV) Figures and Tables that are collapsible/expandable online. A maximum of 5 EV Figures can be typeset. EV Figures should be cited as 'Figure EV1, Figure EV2' etc... in the text and their respective legends should be included in the main text after the legends of regular figures.
 - For the figures that you do NOT wish to display as Expanded View figures, they should be bundled together with their legends in a single PDF file called *Appendix*, which should start with a short Table of Content. Appendix figures should be referred to in the main text as: "Appendix Figure S1, Appendix Figure S2" etc.
 - Additional Tables/Datasets should be labelled and referred to as Table EV1, Dataset EV1, etc. Legends have to be provided in a separate tab in case of .xls files. Alternatively, the legend can be supplied as a separate text file (README) and zipped together with the Table/Dataset file.
- See detailed instructions here: <https://www.embopress.org/page/journal/17574684/authorguide#expandedview>
- Dataset EV legends: Suppl. tables S1 and S3 should be uploaded as separate tables and renamed "Dataset EV1" and "Dataset EV2". The remaining tables should be renamed "Table EV1" etc and uploaded as separate files.
 - Please make sure all figures / figure panels are referenced in the text (callout currently missing for Figure 7D and Suppl. Tables S1 and S2).
 - Please address the queries from our data editors in the figure legends. These queries will be sent to you in a few days.

4/ At EMBO Press we ask authors to provide source data for the main figures. Our source data coordinator will contact you to discuss which figure panels we would need source data for and will also provide you with helpful tips on how to upload and organize the files.

5/ Checklist: please fill in the following sections:

- Cells / cell authentication and mycoplasma contamination
- DNA/RNA sequences
- Experimental study design and statistics
- Sample definition: technical or biological replicates
- Ethics: Please check the section Specimen and field samples, as I'm not sure it applies to your study.

6/ The paper explained: EMBO Molecular Medicine articles are accompanied by a summary of the articles to emphasize the major findings in the paper and their medical implications for the non-specialist reader. Please provide a draft summary of your article highlighting

7/ Every published paper now includes a 'Synopsis' to further enhance discoverability. Synopses are displayed on the journal webpage and are freely accessible to all readers. They include a short stand first (maximum of 300 characters, including space) as well as 2-5 one-sentences bullet points that summarizes the paper. Please write the bullet points to summarize the key NEW findings. They should be designed to be complementary to the abstract - i.e. not repeat the same text. We encourage inclusion of key acronyms and quantitative information (maximum of 30 words / bullet point). Please use the passive voice. Please attach these in a separate file or send them by email, we will incorporate them accordingly.

8/ As part of the EMBO Publications transparent editorial process initiative (see our Editorial at <http://embomolmed.embopress.org/content/2/9/329>), EMBO Molecular Medicine will publish online a Review Process File (RPF) to accompany accepted manuscripts.

This file will be published in conjunction with your paper and will include the anonymous referee reports, your point-by-point response and all pertinent correspondence relating to the manuscript. Let us know whether you agree with the publication of the RPF and as here, if you want to remove or not any figures from it prior to publication.

I look forward to receiving your revised manuscript.

Yours sincerely,

Lise Roth

***** Reviewer's comments *****

Referee #1 (Comments on Novelty/Model System for Author):

The PolyIC injection is clinically still not really relevant. The data of the authors are very good.

Referee #1 (Remarks for Author):

All points clarified. Thank You.

Referee #2 (Comments on Novelty/Model System for Author):

Revision has sufficiently addressed my main concerns.

Referee #2 (Remarks for Author):

The authors have substantially addressed my main concerns except I remain skeptical on expression of TLR3 on CD8+ T cells. From the tissue IF images presented, it seems like TLR3 is expressed everywhere and not specifically on CD8+ T cells. It is also known that main source of TLR3 is more on innate rather than adaptive immune cells. I would suggest to moderate the claim on TLR3 signaling on CD8+ T cells. It will make more sense to suggest the effect is most likely indirect instead.

Referee #3 (Remarks for Author):

I thank the authors for the provided answers and for this interesting work.

Referee #1 (Comments on Novelty/Model System for Author):

The PolyIC injection is clinically still not really relevant. The data of the authors are very good.

Response:

Many thanks for reviewer's recognition of our data and extremely valuable comments for us. To address this, we have supplemented the clinical relevance of the poly(I:C) injection in the "Discussion" section, as follows: "Polyinosinic-polycytidylic acid [Poly(I:C)] injection for clinical use was successfully adopted in 1971. As a synthetic dsRNA, poly(I:C) could activate host defense in a pattern similar to that of viral infection, and regulates and optimizes antigen-specific immune responses as immunomodulators through pathogen-associated molecular patterns by bounding to antigens properly, thereby stimulating cell-mediated immune responses to kill viruses (De Waele *et al.*, 2021). Therefore, it was widely used as a clinical broad-spectrum antiviral drug in China with high safety and very low side effects for half a century, including the treatment of hepatitis B. However, with the rapid development of antiviral drugs, more powerful antiviral drugs significantly superior to poly(I:C) were emerging continuously, therefore the clinical application of poly(I:C) is greatly reduced. Recently, with the rapid development of tumor vaccines, poly(I:C) and its derivatives have been widely used as vaccine adjuvants in various tumor vaccine-related clinical trials around the world (Cai *et al.*, 2021; Le Naour *et al.*, 2020; Ogino *et al.*, 2022), demonstrating a safe and efficient features for activating the body's immune response. In this study, antigens released from irradiated tumor cells were similar to in situ auto-vaccination, and poly(I:C) could enhance the effect of this vaccination through promoting the presentation of antigens released from irradiated tumor cells to recruit activated CD8⁺ T cells to the tumor sites. Although the enrollment of patients is limited in our study, RT combined with poly(I:C) has proven to be a safety and feasible strategy in HCC. Importantly, it is very cheap, with the

price only one-hundredth of that of immune checkpoint inhibitors, making it very suitable for large-scale clinical promotion, especially in developing countries.”

Referee #1 (Remarks for Author):

All points clarified. Thank You.

Response:

Thanks again for reviewer’s recognition of our work.

Referee #2 (Comments on Novelty/Model System for Author):

Revision has sufficiently addressed my main concerns.

Response:

Thanks very much for reviewer’s high approval of our revision. The extremely valuable comments that the reviewer raised had profoundly beneficial to our research.

Referee #2 (Remarks for Author):

The authors have substantially addressed my main concerns except I remain skeptical on expression of TLR3 on CD8⁺ T cells. From the tissue IF images presented, it seems like TLR3 is expressed everywhere and not specifically on CD8⁺ T cells. It is also known that main source of TLR3 is more on innate rather than adaptive immune cells. I would suggest to moderate the claim on TLR3 signaling on CD8⁺ T cells. It will make more sense to suggest the effect is most likely indirect instead.

Response:

We sincerely thanks for reviewer’s professional and meticulous review of our manuscript. We strongly agree with the reviewer's statement that TLR3 expression is not specific on CD8⁺ T cells, but is rather present on a range of cells, which may include various innate immune cells. Following the reviewer suggestion, we have moderated the claim on TLR3 signaling on CD8⁺ T cells. Moreover, we indeed could not dismiss the potential role of poly(I:C) in enhancing abscopal effect through the activation of TLR3 signaling in other innate immune cells, and then potentially leading them to initiate CD8⁺ T cell-driven anti-tumor immune responses. This intriguing possibility is what we aim to investigate in our forthcoming studies. Thanks again for your very enlightening comment.

Referee #3 (Remarks for Author):

I thank the authors for the provided answers and for this interesting work.

Response:

Thanks very much for the reviewer' recognition of our revision and response. your extremely constructive comments on our manuscript have made our work more interesting and meaningful.

9th Apr 2024

Dear Prof. Liu,

Thank you for sending your revised files. I am pleased to inform you that your manuscript is accepted for publication and is now being sent to our publisher to be included in the next available issue of EMBO Molecular Medicine.

Yours sincerely,

Lise Roth
